# Automaton Constrained Q-Learning

**Anastasios Manganaris, Vittorio Giammarino, and Ahmed H. Qureshi**
Department of Computer Science
Purdue University
{amangana,vgiammar,ahqureshi}@purdue.edu

## Abstract

Real-world robotic tasks often require agents to achieve sequences of goals while respecting time-varying safety constraints. However, standard Reinforcement Learning (RL) paradigms are fundamentally limited in these settings. A natural approach to these problems is to combine RL with Linear-time Temporal Logic (LTL), a formal language for specifying complex, temporally extended tasks and safety constraints. Yet, existing RL methods for LTL objectives exhibit poor empirical performance in complex and continuous environments. As a result, no scalable methods support both temporally ordered goals and safety simultaneously, making them ill-suited for realistic robotics scenarios. We propose Automaton Constrained Q-Learning (ACQL), an algorithm that addresses this gap by combining goal-conditioned value learning with automaton-guided reinforcement. ACQL supports most LTL task specifications and leverages their automaton representation to explicitly encode stage-wise goal progression and both stationary and non-stationary safety constraints. We show that ACQL outperforms existing methods across a range of continuous control tasks, including cases where prior methods fail to satisfy either goal-reaching or safety constraints. We further validate its real-world applicability by deploying ACQL on a 6-DOF robotic arm performing a goal-reaching task in a cluttered, cabinet-like space with safety constraints. Our results demonstrate that ACQL is a robust and scalable solution for learning robotic behaviors according to rich temporal specifications.

## 1 Introduction

Achieving desirable robot behavior in real-world applications often requires managing long and complex sequences of subgoals while adhering to strict safety constraints. For instance, autonomous mobile robots in warehouse settings must navigate to restock shelves, all while avoiding obstacles and remaining within a reachable distance from the recharging station for their current battery level. Similarly, a nursing robot must routinely complete its rounds in a given sequence, unless notified of an emergency situation.

Despite the success of Goal-Conditioned Reinforcement Learning (GCRL) [1, 2] and Safe RL [3, 4], these approaches fall short in addressing the combined challenges of sequential goals and dynamic safety constraints. GCRL is effective for reaching individual goals but lacks mechanisms for reasoning over goal sequences or ensuring safety. Safe RL enforces fixed safety constraints, but typically assumes they remain static throughout the task, limiting its applicability to temporally evolving tasks.

Formal languages such as LTL [5] provide an expressive framework for specifying complex tasks involving multiple interdependent goals and non-stationary safety constraints. However, reward functions defined directly from LTL specifications are inherently non-Markovian, since temporal properties refer to entire trajectories rather than individual transitions. This violates the Markov assumption—a fundamental premise of standard RL—which assumes that transitions and rewards

depend only on the current state and action. As a result, RL algorithms built on the Markov Decision Process (MDP) formalism [6, 7] are ill-suited for direct application to LTL-defined objectives.

To overcome this, some studies [8–10] have proposed to translate the LTL formula into an automaton that tracks temporal progress toward task completion and enables defining Markovian rewards corresponding to temporal specifications. While this approach allows for potentially handling arbitrary LTL formulas, its generality is tied to the use of sparse binary rewards derived from the Boolean evaluation of the specification. In complex environments, however, such sparse signals are encountered too infrequently to effectively guide behavior, and designing denser rewards typically requires additional domain knowledge not readily available in many robotics tasks. Furthermore, these methods only implicitly deal with ensuring safety, as safety properties are fundamental components of many LTL specifications [11], and typically rely on ad-hoc mechanisms, such as halting rollouts [12] and reward shaping [13], both of which tend to be brittle and ineffective in complex domains. Other approaches seek to improve scalability by employing hierarchical [13] or goal-conditioned [14] policies but sacrifice generality, particularly with respect to safety constraints.

To bridge the gap between LTL-capable algorithms that scale poorly and scalable RL algorithms that lack support for most LTL tasks, we propose ACQL, which lifts Safe RL and GCRL to the class of problems expressible as LTL formulae in the recurrence class [15]. This algorithm, which we consider our primary contribution, is built on top of two technical novelties that address distinct challenges associated with LTL problems. First, to overcome the poor scalability of sparse rewards, we encode automaton states with their associated goals, enabling the use of goal-conditioned techniques such as Hindsight Experience Replay (HER) [2] to densify reward signals. Second, inspired by Hamilton-Jacobi (HJ) reachability analysis [16, 17], we employ a minimum-safety-based product Constrained Markov Decision Process (CMDP) formulation that enforces compliance with arbitrary LTL safety constraints at optimality. We demonstrate these technical contributions are necessary in enabling our ACQL algorithm to significantly outperform other algorithms for solving LTL tasks. Additionally, we demonstrate its effectiveness in learning real-world-deployable policies for a 6-DOF robot arm operating in a storage cabinet environment.

## 2   Related Work

The use of LTL specifications with RL algorithms has been a popular topic in recent years [10, 13, 18–28]. One approach to this topic has been directly defining non-Markovian reward functions based on overall task satisfaction, but this only supports LTL formulae that are satisfiable over finite prefixes [18, 29–31]. In restricted cases where task satisfaction is differentiable with respect to a policy's actions, it is also possible to directly train sequence models that satisfy the specification [23]. The most popular and general methods, however, convert LTL tasks into automata to use in conjunction with standard deep RL techniques for MDPs [8–10, 19, 22, 24–28, 32–36]. Within these works, there are two subcategories including methods for learning policies that can perform well across a set of multiple LTL tasks [20, 22, 25, 27] and methods that focus on optimally satisfying a single LTL task [8–10, 12, 19, 28, 32, 36, 37].We specifically aim to improve on the scalability of these latter methods in high-dimensional environments. Of these techniques, we specifically highlight Reward Machines (RMs) [37] as a framework with similar generality to our method and the first baseline we compare against.

Our work utilizes techniques from GCRL [38, 39] to accelerate learning for LTL tasks. Other methods have taken a related approach by learning or re-using goal-conditioned policies that are hierarchically guided by LTL expressions to enable zero-shot generalization to new tasks [14, 22, 26, 27]. Using discrete skills as opposed to a goal-conditioned policy has also been explored in [13, 21]. All these approaches require the user to train a different skill for every automaton edge or a different goal-conditioned policy for every safety constraint that appears in the task, except for limited classes of safety constraints. This is impractical for most real-world tasks and, in particular, when safety constraints are changing throughout a task. Conversely, our method learns a single goal-conditioned policy for all goals in a particular LTL expression while still accounting for arbitrary, potentially non-stationary safety constraints. While ACQL has been developed for online RL rather than zero-shot generalization, there is a significant overlap in the tasks that both can ultimately solve. Therefore, we include the Logical Options Framework (LOF) [13] as a representative for these methods in our experiments.

Our work is also focused on solving LTL tasks that feature safety constraints. Safety constraints are often indirectly addressed in the methods discussed above. They primarily focus on making unsafe actions suboptimal by either terminating rollouts [8, 12, 36, 37] or applying negative reward shaping [13, 20, 32] when an unsafe automaton transition occurs. Other approaches based on formal methods, such as shielding [40, 41] and runtime model-checking [42], can more robustly enforce safety requirements but typically rely on additional assumptions such as access to a discrete abstraction of the environment or known system dynamics. In contrast, techniques from the Safe RL literature can still robustly enforce stationary safety constraints in complex environments without such assumptions [3, 4, 43–47]. Our approach builds on these Safe RL methods to address non-stationary safety constraints induced by LTL specifications.

## 3 Preliminaries

**Constrained Reinforcement Learning**   RL problems with constraints are typically modeled as discounted CMDPs, which are defined by a tuple $(\mathcal{S}, \mathcal{A}, \mathcal{T}, d_0, r, c, \mathcal{L}, \gamma)$. This tuple consists of a state space $\mathcal{S}$, an action space $\mathcal{A}$, a transition function $\mathcal{T} : \mathcal{S} \times \mathcal{A} \to P(\mathcal{S})$ (where $P(\mathcal{S})$ represents the space of probability measures over $\mathcal{S}$), an initial state distribution $d_0 \in P(\mathcal{S})$, reward and constraint functions $r$, $c : \mathcal{S} \times \mathcal{A} \to \mathbb{R}$, a discount factor $\gamma$, and a limit for constraint violation $\mathcal{L} \in \mathbb{R}$ [3]. Given a CMDP, the standard objective is to find the stationary policy $\pi : \mathcal{S} \to P(\mathcal{A})$ satisfying the constrained maximization

$$\max_{\pi} \ J_r(\pi) \quad \text{s.t.} \quad J_c(\pi) < \mathcal{L}, \tag{1}$$

where $J_r(\pi) = \mathbb{E}_{\tau \sim \pi} \left[ \sum_{t=0}^{\infty} \gamma^t r(s_t, a_t) \right]$ and $J_c(\pi) = \mathbb{E}_{\tau \sim \pi} \left[ \sum_{t=0}^{\infty} \gamma^t c(s_t, a_t) \right]$ are respectively the expected total discounted return and cost of the policy $\pi$ over trajectories $\tau = (s_0, a_0, s_1, a_1, \dots)$ induced by $\pi$ through interactions with the environment of the CMDP. We denote the state value function of $\pi$ by $V_{\pi}^r(s) = \mathbb{E}_{\tau}[\sum_{t=0}^{\infty} \gamma^t r(s_t, a_t) | s_0 = s]$ and the state-action value function by $Q_{\pi}^r(s, a) = \mathbb{E}_{\tau}[\sum_{t=0}^{\infty} \gamma^t r(s_t, a_t) | s_0 = s, a_0 = a]$.

**Temporal Logic**   LTL [5] is an extension of propositional logic for reasoning about systems over time. LTL formulae $\phi \in \Phi$ consist of atomic propositions $p$ from a set $AP$, the standard boolean operators "not" ($\neg$), "and" ($\wedge$), and "or" ($\vee$), and temporal operators that reference the value of propositions in the future. These operators are "next" ($\circ$), "eventually" ($\diamond$), and "always" ($\square$). We follow the definitions for these operators given in [48]. In our algorithm, we use Signal Temporal Logic (STL), which further extends LTL with quantitative semantics and requires that every atomic proposition $p \in AP$ is defined as a real-valued function of the system state. The quantitative semantics is defined by a function $\rho : \mathcal{S}^{\omega} \times \Phi \to \mathbb{R}$ that produces a robustness value representing how much a sequence of states $\omega \in \mathcal{S}^{\omega}$ satisfies or violates the property specified by an STL formula $\phi \in \Phi$. Our exact implementation of these quantitative semantics is based on [49]. Every temporal logic constraint can be expressed as the conjunction of a "safety" and "liveness" constraint [50], where a safety constraint is informally defined as a requirement that something must never happen for the task to succeed, and a liveness constraint is defined as a requirement that something must happen for the task to succeed. We take advantage of this dichotomy in our method.

**Automata**   The robotics tasks discussed so far can be expressed more specifically as STL formulae in the recurrence class, as defined in [15], and all such formulae can be translated into abstract machines called Deterministic Büchi Automata (DBAs) [48]. A DBA is formally defined by the tuple $A = (\Sigma, \mathcal{Q}, \delta, q_0, F)$, consisting of an alphabet $\Sigma$, a set of internal states $\mathcal{Q}$, a transition function $\delta : \mathcal{Q} \times \Sigma \to \mathcal{Q}$, an initial state $q_0 \in \mathcal{Q}$, and a set of accepting states $F \subseteq \mathcal{Q}$. An automaton is used to process an infinite sequence of arbitrary symbols from the alphabet $\Sigma$ and determine whether the sequence satisfies or does not satisfy the original logical expression. In our setting, a symbol processed by the automaton is a subset of atomic propositions $l \in 2^{AP}$ that are true in some MDP state $s \in \mathcal{S}$. This symbol $l$ is referred to as the state's labeling and the mapping of states to their labeling is denoted by a labeling function $L : \mathcal{S} \to 2^{AP}$. The automaton $A$ is always in some state $q \in \mathcal{Q}$, and starts in the state $q_0$. Each element $\sigma \in \Sigma$ of an input sequence causes some change in the automaton's internal state according to the transition function $\delta$. For convenience, we will refer to edges in the automaton with transition predicates. A transition predicate is a propositional formula that holds for all $\sigma \in \Sigma$ which induce a transition between two states $q_i, q_j \in \mathcal{Q}$ according to $\delta$ [11]. By processing each state while an agent interacts with an MDP, the internal automaton state provides

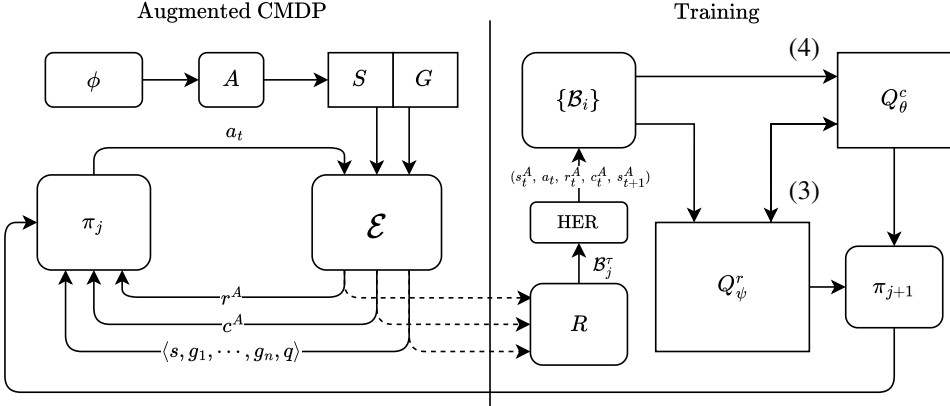

Figure 1: The ACQL algorithm relies on a novel augmented formulation of CMDPs (left). An input task specification $\phi$ is converted into the DBA $A$, from which safety constraints and subgoals are collected into mappings $S$ and $G$ respectively. The learning agent receives subgoals $g_1, \cdots, g_n$ at every stage in the task from $G$ and safety constraint feedback in $c^A$ from $S$. Trajectories induced by the policy $\pi_j$ are collected into a replay buffer $R$, from which batches $\mathcal{B}_j^\tau$ are sampled and modified using HER [2]. From these modified trajectories, mini-batches $\mathcal{B}_i$ of transitions $(s_t^A, a_t, r_t^A, c_t^A, s_{t+1}^A)$ are used to compute the targets $y_t^r$ in (3) and $y_t^c$ in (4) for training models of the state-action value function and safety function, $Q_\theta^r$ and $Q_\psi^c$, from which an updated policy $\pi_{j+1}$ is derived.

information to the agent about its current progress in the task. This is the fundamental idea behind a Product MDP, which combines any MDP with a deterministic automaton and is a standard technique in model checking for probabilistic systems [48]. A complete definition for a product MDP in an RL context can be found in previous work such as [10, 37].

## 4 Method

This section introduces our ACQL algorithm along with a novel augmented product CMDP on which it is based. An overview is provided in Figure 1. First, we detail the construction of the augmented product CMDP derived from a task specification. Observations in this CMDP include the set of subgoals associated with the current automaton state $q$, as described in Section 3. This CMDP also uses safety conditions obtained from the task automaton to provide separate constraint feedback that the learning agent uses to determine safe actions. Unlike the standard CMDP constraint in (1), we argue for the use of a constraint on the minimum safety that is easier to learn. We finally present an overview and brief analysis of ACQL, which is tailored to utilize the information provided by our augmented product CMDP to scale beyond existing algorithms for learning in product MDPs.

### 4.1 Augmented Product CMDP Formulation

**Obtaining the Automaton, Safety, and Liveness Constraints**    To construct the augmented product CMDP used by our algorithm, we begin by translating the input STL task specification $\phi$ into a DBA $A = (\Sigma, \mathcal{Q}, \delta, q_0, F)$ using the SPOT library [51]. As described in Section 3, this automaton represents the structure of the task. To better guide learning, we extract from this automaton additional structure that captures the task's safety and liveness requirements. Intuitively, these represent conditions that must *always* hold to avoid failure (safety) and conditions that must *eventually* hold to make progress toward task completion (liveness) [11, 50]. This information is summarized in two mappings: a safety condition mapping $S : \mathcal{Q} \to \Phi$, which assigns to each automaton state a proposition that must remain true, and a liveness condition mapping $O : \mathcal{Q} \to \Phi$, which assigns to each automaton state a proposition that must be eventually satisfied to proceed in the task. Based on $O$, we define $G : \mathcal{Q} \to \mathcal{G}^+$ to explicitly link automaton states to subgoals, where $\mathcal{G}^+ = \bigcup_{i=1}^n \mathcal{G}^i$

represents the full space of possible subgoal lists up to a task-dependent length $n$ and $\mathcal{G}^i$ is the $i$-fold cartesian product of $\mathcal{G}$ with itself.

First, to obtain safety constraints, we determine the non-accepting sink-components of the automaton $A$; i.e., the automaton states $Q \setminus F$ from which there is no path to any state in $F$. For each automaton state $q$, we then examine the outgoing transitions from $q$ that lead into these sink components. The predicates guarding these transitions describe conditions under which a transition into an unsafe state would occur, so negating these predicates provides the conditions that must hold in order to stay safe while in state $q$. We can then define $S(q)$ as the conjunction of all such negated predicates for state $q$.

Second, to obtain liveness constraints, we ignore the above transitions for safety conditions and examine the remaining predicates guarding outgoing transitions from each state $q$. These can only specify the conditions necessary to make progress in the task. Therefore, we can take the disjunction of these transition predicates to obtain the formula $O(q)$ that must be satisfied in order to transition beyond $q$. Following prior work on learning LTL tasks [13, 14], we assume that a subset of the atomic propositions, $AP_{\text{subgoal}} \subseteq AP$, represent subgoal propositions and are parameterized by values $g$ within a subgoal space $\mathcal{G} \subseteq \mathcal{S}$. Note that we consider only a subset since not all the propositions in our modified DBA are necessarily related to achieving subgoals. Using this fact, we filter the predicates in $O(q)$ to retain only those associated with these subgoal propositions, which in turn provides a list of subgoals $G(q) = g_1, \ldots, g_n$ that are relevant to the logical formula given by $O(q)$. An illustrative example for these two steps is provided in the appendix.

**Defining the Augmented Product CMDP** Now, let $\mathcal{M}^A = (\mathcal{S}^A, \mathcal{A}^A, \mathcal{T}^A, d_0^A, r^A, c^A, \gamma, \mathcal{L})$ be a new CMDP formed by augmenting the original MDP $\mathcal{M} = (\mathcal{S}, \mathcal{A}, \mathcal{T}, d_0, r, \gamma)$ with the DBA $A = (\Sigma, \mathcal{Q}, \delta, q_0, F)$. The new state space $\mathcal{S}^A = \mathcal{S} \times \mathcal{G}^+ \times \mathcal{Q}$ is the Cartesian product of the original state space $\mathcal{S}$, the space for every possible list of subgoals $\mathcal{G}^+$, and the set of automaton states $\mathcal{Q}$. A state $s^A \in \mathcal{S}^A$ can be written as $\langle s, g^+, q \rangle$, where $s \in \mathcal{S}$, $g^+ \in \mathcal{G}^+$, and $q \in \mathcal{Q}$ are the constituent MDP state, goal-list and automaton state of $s^A$. The action space $\mathcal{A}^A = \mathcal{A}$ is unmodified. The new transition dynamics $\mathcal{T}^A$ are defined so that a transition to $\langle s', g^{+\prime}, q' \rangle \in \mathcal{S}^A$ from $\langle s, g^+, q \rangle \in \mathcal{S}^A$ is impossible if the automaton does not support a transition from $q$ to $q'$ when entering the state $s'$; i.e.,

$$\mathcal{T}^A(\langle s', g^{+\prime}, q' \rangle | \langle s, g^+, q \rangle, a) = \begin{cases} \mathcal{T}(s'|s, a) & \text{if } q' = \delta(q, L(s')), g^{+\prime} = G(q'), \\ 0 & \text{otherwise.} \end{cases}$$

Likewise, the initial state distribution $d_0^A(\langle s, g^+, q \rangle) = d_0(s)$ if $q = q_0$ and $g^+ = G(q_0)$, and is zero otherwise. The reward function $r^A(\langle s, g^+, q \rangle) = \mathbb{1}_F(q)$, where $\mathbb{1}(\cdot)$ is an indicator function, is defined to provide a sparse reward of 1 when the task is finished and the agent is near the goal associated with any accepting state of the automaton. Reward sparsity was a central shortcoming of prior work, often resulting in poor performance in realistic scenarios. However, our augmented product CMDP specifically includes the subgoal list $g^+$ to facilitate the use of modern GCRL algorithms to mitigate this sparsity in many relevant tasks. In particular, ACQL retroactively assigns rewards based on achieved subgoals using HER, and the benefit of this strategy for LTL-specified tasks is validated in our ablations (Section 5.3). Lastly, we define the constraint function $c^A$ along with the constrained objective used with our CMDP formulation.

**Minimum LTL Safety Constraint** Although the sum-of-costs formulation in (1) conveniently leverages the standard Bellman equation, it presents practical challenges in learning. Specifically, predicting the cumulative sum of future costs requires accurately modeling long-term dependencies, which increases variance and slows down convergence. This can be appropriate for maximizing reward but unnecessarily complicated for safety. When dealing with safety. and in particular hard-safety [52], we are interesting in knowing whether the generated trajectory becomes unsafe at least once. Hence, this can be reformulated as a classification task. This approach bypasses the need for accurate regression by directly learning the decision boundary for state-action pairs leading to safety violations. To achieve this, we define a constraint function $c^A$ such that it takes negative values if a safety constraint is violated and positive values otherwise, and we apply a constraint based on the minimum over all future values of $c^A(s_t^A, a_t)$. Then, we simply learn to classify this minimum value as being either positive or negative. This formulation allows us to determine safety violations in a single step, avoiding the complexity of cumulative cost prediction. A major advantage of this formulation is that it removes the need for manual tuning of the violation limit $\mathcal{L} \in \mathbb{R}$, which depends on the range of possible costs. Instead, by directly classifying actions as leading to a safety violation,

---

**Algorithm 1** Automaton Constrained Q-Learning

---

**Require:** An MDP $\mathcal{M} = (\mathcal{S}, \mathcal{A}, \mathcal{T}, d_0, r, \gamma)$, an STL specification $\phi \in \Phi$, a safety limit $\mathcal{L} \in [-1, 1]$, a learning rate $\alpha$, an interpolation factor $\lambda$

1:   $A \leftarrow \text{TRANSLATE}(\phi)$
2:   $S, G \leftarrow \text{PARTITION}(A)$
3:   $\mathcal{M}^A \leftarrow (\mathcal{S}^A, \mathcal{A}, \mathcal{T}^A, d_0^A, r^A, c^A, \gamma, \mathcal{L})$
4:   $Q_\theta^c, Q_\psi^r \leftarrow \text{MAKENETWORKS}(\mathcal{S}^A, \mathcal{A})$
5:   $\bar{\theta} \leftarrow \theta, \bar{\psi} \leftarrow \psi$
6:   $R \leftarrow \text{MAKEREPLAYBUFFER}(\mathcal{M}^A)$
7:   **for** $j = 1, \ldots, N$ **do**
8:      $\gamma_c \leftarrow \text{SAFETYGAMMASCHEDULER}(j)$
9:      $\pi_j(s^A) \leftarrow \arg\max_{a: Q_\theta^c(s^A, a) > \mathcal{L}} Q_\psi^r(s^A, a)$
10:     $\tau \leftarrow \text{GETTRAJECTORY}(\mathcal{M}^A, \pi_j)$
11:     $R \leftarrow R \cup \tau$.
12:     $\mathcal{B}_j^\tau \sim R$
13:     $\mathcal{B}_j^\tau \leftarrow \text{RELABEL}(\mathcal{B}_j^\tau)$
14:     **for** $i = 1, \ldots, M$ **do**
15:        $\mathcal{B}_i \leftarrow \{(s_t^A, a_t, r_t^A, c_t^A, s_{t+1}^A)\} \sim \mathcal{B}_j^\tau$
16:        $\theta \leftarrow \theta - \alpha \nabla_\theta L_{j,i}^c(\theta)$
17:        $\psi \leftarrow \psi - \alpha \nabla_\psi L_{j,i}^r(\psi)$
18:        $\bar{\theta} \leftarrow (1 - \lambda)\bar{\theta} + \lambda\theta, \bar{\psi} \leftarrow (1 - \lambda)\bar{\psi} + \lambda\psi$
19:     **end for**
20: **end for**

---

we reduce the range of $\mathcal{L}$ to $[-1, 1]$. This normalization ensures robustness across tasks and allows for a simple, task-agnostic choice of $\mathcal{L} = 0$. Directly defining $c^A(\langle s, g^+, q \rangle, a) = \rho(s, S(q))$ using the robustness function for $S(q)$ (see Section 3) satisfies this formulation and corresponds to the safety constraints for any STL-specified task.

Now, we can define our new objective as finding the optimal stationary policy $\pi^* : \mathcal{S} \to P(\mathcal{A})$ that maximizes the expected total discounted return, as defined in (1), while keeping the expected minimum safety above the safety limit $\mathcal{L}$:

$$\mathbb{E}_{\tau \sim \pi} \left[ \min_{t \in [0, \infty]} c^A(s_t, a_t) \right] > \mathcal{L}, \tag{2}$$

where $\tau = (s_0^A, a_1, s_2^A, a_2, \dots)$ is a trajectory within our augmented CMDP induced by the policy $\pi$. We denote this expected minimum safety for a policy $\pi$ by its state-action safety function $Q_\pi^c(s, a) = \mathbb{E}_{\tau \sim \pi}[\min_{t=0}^\infty c^A(s_t^A, a_t) | s_0^A = s, a_0 = a]$.

### 4.2 Automaton Constrained Q-Learning (ACQL)

**Overview**   In what follows, we provide an overview of the ACQL algorithm, whose pseudocode is shown in Algorithm 1. First, the input STL specification $\phi$ is translated into an automaton $A$ (Line 1) that is partitioned to produce the mappings $S$ and $G$ (Line 2). These mappings and the automaton are combined with the input MDP $\mathcal{M}$ to create the augmented CMDP in Line 3. Using the augmented CMDP, ACQL can proceed to learn the optimal policy

$$\pi^*(s^A) = \underset{a \,:\, Q^{c*}(s^A, a) > \mathcal{L}}{\arg\max} Q^{r*}(s^A, a),$$

by learning the optimal state-action value function $Q^{r*} = Q_{\pi^*}^r$ and the optimal state-action safety function $Q^{c*} = Q_{\pi^*}^c$. In order to learn these optimal functions, we define the models $Q_\psi^r : \mathcal{S}^A \times \mathcal{A} \to \mathbb{R}$ and $Q_\theta^c : \mathcal{S}^A \times \mathcal{A} \to [-1, 1]$ parameterized by $\psi$ and $\theta$ (Line 4). Their initial parameters are copied to initialize the target parameters $\bar{\theta}$ and $\bar{\psi}$ in Line 5, and an empty replay buffer is initialized in Line 6.

The algorithm proceeds to iterate for $N$ epochs indexed by $j$. At each epoch, we obtain a value for the safety discount factor $\gamma_c$ in Equation (4), which asymptotically approaches 1.0 as training progresses

(Line 8). The policy $\pi_j$ for the epoch is defined to select the most-rewarding action according to $Q^r_\psi$ constrained by $Q^c_\theta$ (Line 9). A trajectory $\tau$ is collected according to an epsilon-greedy version of this policy (Line 10), and this trajectory is added to the replay buffer (Line 11). From this replay buffer, a batch of trajectories $\mathcal{B}^\tau_j$ is sampled every epoch (Line 12). The inclusion of subgoals in the states $s^A_t$ of the sampled mini-batch $\mathcal{B}_i$ allows us to further accelerate learning using relabeling techniques such as HER [2], which allows $Q^r_\theta$ to improve from failed attempts at progressing through the task. Mini-batches of transitions $\mathcal{B}_i$ are subsequently sampled from the relabeled trajectories (Line 15) and used to update $Q^r_\psi$ for minimizing the loss

$$L^r_{j,i}(\psi) = \mathop{\mathbb{E}}_{(s^A_t, a_t) \sim \mathcal{B}_i} \left[ \left( y^r_t - Q^r_\psi(s^A_t, a_t) \right)^2 \right], \quad \text{with} \quad y^r_t = r^A_t + \gamma Q^r_{\bar\psi}(s^A_{t+1}, \pi_j(s^A_{t+1})). \quad (3)$$

Similarly, $Q^c_\theta$ is trained with the target

$$y^c_t = \gamma_c \min\{c^A(s^A_t, a_t), \, Q^c_{\bar\theta}(s^A_{t+1}, \pi_j(s^A_{t+1}))\} + (1 - \gamma_c) \, c^A(s^A_t, a_t), \quad (4)$$

which is derived from the Bellman principle of optimality for expected minimum cost objectives given in [16]. The targets, $y^r$ and $y^c$, are computed using target parameters, $\bar\psi$ and $\bar\theta$, which are updated towards their corresponding main parameters each step with an interpolation factor $\lambda$ (Line 18). Crucially, we schedule the discount factor $\gamma_c$ throughout training to asymptotically approach 1.0, which is necessary for convergence to $Q^{c*}$. Additional details for the subroutines used in Algorithm 1 are provided in the appendix.

**Analysis**   Under mild assumptions, ACQL is guaranteed to asymptotically converge to the optimal solution. We summarize this in the following proposition:

**Proposition 1.** *Let $\mathcal{M}^A$ be an augmented CMDP with $|\mathcal{S}^A| < \infty$, $|\mathcal{A}| < \infty$, and $\gamma \in [0, 1)$, and let $Q^c_n$ and $Q^r_n$ be models for the state-action safety and value functions indexed by $n$. Assume they are updated using Robbins-Monro step sizes $a(n)$ and $b(n)$, respectively, with $b(n) \in o(a(n))$ according to (3) and (4). Assume that $\gamma_{c_n}$ is also updated with step sizes $c(n)$ such that $\gamma_{c_n} \to 1$ and $c(n) \in o(b(n))$. Then $Q^c_n$ and $Q^r_n$ converge to $Q^{c*}$ and $Q^{r*}$ almost surely as $n \to \infty$.*

*Proof Sketch.* The step sizes $a(n)$, $b(n)$, and $c(n)$ create a stochastic approximation algorithm on three timescales [53]. Based on the contraction property of (4), the fastest updating $Q^c_n$ asymptotically tracks the correct safety function for the policy determined by $Q^r_n$ and $\gamma_{c_n}$. On the slower timescale ensured by the definition of $b(n)$ and $a(n)$, $Q^r_n$ converges to the fixed point as determined by the relatively static $\gamma_{c_n}$, due to the contraction property of (3). Finally, as $\gamma_c$ converges on the slowest timescale, the fixed points for the two faster timescales approach $Q^{r*}$ and $Q^{c*}$ at $\gamma_c = 1$.   $\square$

In Proposition 1, the notation $b(n) \in o(a(n))$ means that $b(n)/a(n) \to 0$ asymptotically. The assumption of a finite MDP is standard in the literature, but relaxing this assumption is possible by applying fixed point theorems for infinite dimensional spaces [54]. After convergence, the final policy will behave optimally for the reward $r$ and (if possible) never incur any cost, thereby respecting the LTL safety constraint. A complete description of our algorithm, model implementation details, and a full proof of convergence, with reference to similar arguments used in prior work [43, 55, 56], are provided in the appendix.

## 5   Experiments

In this section, we justify the design of ACQL and evaluate its effectiveness. We conducted a comparative analysis of ACQL against established baselines in RL from LTL specifications, including RMs [12] and the LOF [13]. We demonstrate our method's real-world applicability by solving more complex LTL tasks with a 6-DOF robot arm in a storage cabinet environment. Lastly, we performed an ablation study to clarify how our subgoal-including product CMDP, in combination with HER [2], and our minimum-safety constraint formulation contribute to ACQL's performance.

### 5.1   Comparative Analysis

**Baselines**   In selecting our baselines, we chose to not compare against standard Safe RL methods, offline policy learning methods, and multi-task LTL methods. Safe RL methods, without significant

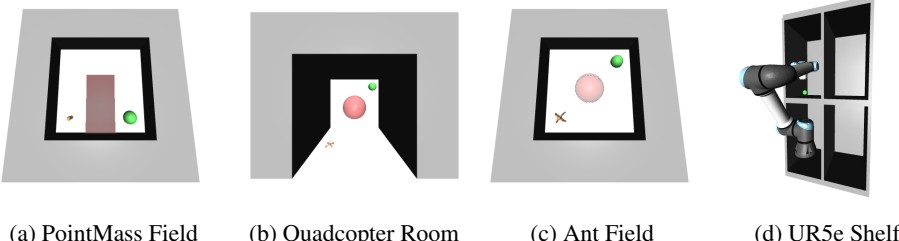

    (a) PointMass Field      (b) Quadcopter Room      (c) Ant Field      (d) UR5e Shelf

Figure 2: We conducted experiments in simulated environments for navigation tasks using a point mass, quadcopter, and ant quadruped. We also trained policies for end-effector control of a UR5e manipulator in a simulated shelf-environment for our real-world experiments.

Table 1: Results from policies trained for five task types in three different environments with 5 million environment interactions and 5 seeds. We report mean and one standard deviation of total reward and success rate in 16 evaluation episodes lasting 1000 steps over seeds for policies at the end of training.

| | | ACQL (Ours) | | LOF [13] | | CRM-RS [37] | |
|---|---|---|---|---|---|---|---|
| Robot | Task | Reward ↑ | S.R.(%) ↑ | Reward ↑ | S.R.(%) ↑ | Reward ↑ | S.R.(%) ↑ |
| P.M. | $\diamond(g_1 \wedge \circ(\diamond g_2))$ | **829.6 ± 9.6** | **83.8 ± 26.4** | 11.0 ± 3.1 | 98.8 ± 2.8 | 91.2 ± 4.3 | 0.0 ± 0.0 |
| | $\diamond g_1 \wedge \diamond g_2$ | **841.1 ± 54.1** | **98.8 ± 2.8** | 470.7 ± 397.5 | 100.0 ± 0.0 | 420.6 ± 259.2 | 61.3 ± 38.6 |
| | $\diamond g_1 \wedge \Box \neg o_1$ | **858.7 ± 3.2** | **100.0 ± 0.0** | 0.0 ± 0.0 | 0.0 ± 0.0 | 0.0 ± 0.0 | 3.8 ± 8.4 |
| | $\neg o_1 \mathcal{U} g_1 \wedge \circ \diamond g_2$ | **525.8 ± 480.0** | **60.0 ± 54.8** | 0.0 ± 0.0 | 0.0 ± 0.0 | 0.0 ± 0.0 | 2.5 ± 5.6 |
| | $\Box\diamond(g_1 \wedge \circ\diamond g_2) \wedge \Box\neg o_1$ | **2.2 ± 2.0** | **62.5 ± 41.8** | 0.0 ± 0.0 | 0.0 ± 0.0 | 0.0 ± 0.0 | 0.0 ± 0.0 |
| Q.C. | $\diamond(g_1 \wedge \circ(\diamond g_2))$ | **813.5 ± 9.7** | **100.0 ± 0.0** | 32.0 ± 6.3 | 98.8 ± 2.8 | 0.0 ± 0.0 | 0.0 ± 0.0 |
| | $\diamond g_1 \wedge \diamond g_2$ | **752.6 ± 157.0** | **92.5 ± 16.8** | 161.7 ± 287.6 | 92.5 ± 16.8 | 0.0 ± 0.0 | 0.0 ± 0.0 |
| | $\diamond g_1 \wedge \Box \neg o_1$ | **822.7 ± 107.4** | **95.0 ± 11.2** | 3.5 ± 5.9 | 15.0 ± 20.5 | 4.1 ± 9.2 | 2.5 ± 5.6 |
| | $\neg o_1 \mathcal{U} g_1 \wedge \circ \diamond g_2$ | **726.6 ± 88.6** | **97.5 ± 3.4** | 0.0 ± 0.0 | 0.0 ± 0.0 | 0.0 ± 0.0 | 0.0 ± 0.0 |
| | $\Box\diamond(g_1 \wedge \circ\diamond g_2) \wedge \Box\neg o_1$ | **2.4 ± 1.2** | **82.5 ± 35.0** | 0.0 ± 0.0 | 0.0 ± 0.0 | 0.0 ± 0.0 | 0.0 ± 0.0 |
| Ant | $\diamond(g_1 \wedge \circ(\diamond g_2))$ | **555.8 ± 39.8** | **98.8 ± 2.8** | 53.2 ± 29.8 | 91.2 ± 9.5 | 5.5 ± 12.3 | 1.2 ± 2.8 |
| | $\diamond g_1 \wedge \diamond g_2$ | **587.2 ± 19.3** | **98.8 ± 2.8** | 158.0 ± 180.6 | 76.2 ± 42.7 | 4.6 ± 6.4 | 2.5 ± 3.4 |
| | $\diamond g_1 \wedge \Box \neg o_1$ | **683.7 ± 31.1** | **85.0 ± 3.4** | 38.3 ± 28.8 | 8.8 ± 7.1 | 0.0 ± 0.0 | 3.8 ± 3.4 |
| | $\neg o_1 \mathcal{U} g_1 \wedge \circ \diamond g_2$ | **193.4 ± 36.2** | **87.5 ± 10.8** | 0.0 ± 0.0 | 0.0 ± 0.0 | 0.0 ± 0.0 | 0.0 ± 0.0 |
| | $\Box\diamond(g_1 \wedge \circ\diamond g_2) \wedge \Box\neg o_1$ | **0.9 ± 0.6** | **77.5 ± 12.9** | 0.0 ± 0.0 | 0.0 ± 0.0 | 0.0 ± 0.0 | 0.0 ± 0.0 |

modification, cannot directly apply to LTL tasks and instead only address tasks with stationary safety constraints. Offline methods assume access to a fixed dataset and are not designed for the online learning setting we address. Multi-task methods [20, 22, 25, 27] take special measures to generalize across a distribution of LTL specifications. Comparisons to these methods in a single-task context would therefore require removing significant components from them and would not yield strong conclusions regarding their relative merit. We instead compared our algorithm against two methods for online RL with singular LTL specifications: Counterfactual Experiences for Reward Machines with Reward Shaping (CRM-RS) [37] and the LOF [13]. CRM-RS employs an RM to define the task and dispense shaped rewards that incentivize task completion. RM states in non-accepting sink components are treated as terminal states, implementing a rollout-terminating strategy to enforce safety constraints. The LOF is a framework for learning hierarchical policies to satisfy LTL tasks. It learns a separate policy, referred to as a logical option, for satisfying every subgoal-proposition in the task. When training each logical option, each step that violates the task's safety propositions receives a large reward penalty (set to $R_s = -1000$ based on their implementation) to discourage unsafe behavior. These policies are then composed with a higher-level policy obtained using value iteration over a discrete state space derived from the task automaton and all the subgoal states of the environment. Both these approaches can handle general LTL specifications and, to our knowledge, represent the most relevant baselines to compare our algorithm against. Furthermore, we note that RMs and the LOF cannot be easily enhanced with high-performing GCRL and Safe RL techniques without adopting substantial modifications to their learning algorithms and product MDP formulation. Our method effectively integrates these components in a single framework that enables learning from general logical objectives in complex settings, as demonstrated in the following tasks.

**Tasks**    We chose five distinct LTL tasks and three different agents to facilitate a thorough comparison between our algorithm and the two baseline methods. All environment simulation was done within the Brax physics simulator [57]. The agents used in our experiments are a 2D PointMass, a Quadcopter, and an 8-DOF Ant quadruped. Additional details for these environments are given in the appendix. We used open environments without physical obstacles for our experiments. Instead, obstacles were

introduced solely through the task specification. This was done to demonstrate the effectiveness of our algorithm in avoiding obstacles based purely on feedback from the task specifications, without confounding effects such as physically restricted motion. The LTL tasks used in the evaluation include: (1) a two-subgoal sequential navigation task between opposite corners of the environment $(\Diamond(g_1 \wedge \circ(\Diamond g_2)))$, (2) a two-subgoal branching navigation task between opposite corners of the environment $(\Diamond g_1 \wedge \Diamond g_2)$, (3) a single-goal navigation task constrained by an unsafe region $(\Diamond g_1 \wedge \Box \neg o_1)$, (4) a two-subgoal navigation task with a disappearing safety constraint $(\neg o_1 \mathcal{U} g_1 \wedge \circ \Diamond g_2)$, and (5) an infinitely-looping navigation task with a persistent safety constraint $(\Box \Diamond (g_1 \wedge \circ \Diamond g_2) \wedge \Box \neg o_1)$. The starting position, obstacle and subgoal configurations for each environment are shown in Figure 2.

**Results** The results for our three simulated environments and five task types are shown in Table 1. All results are reported for the final policy obtained after 5 million environment interactions with five different seeds. For LOF, each individual logical option was afforded its own 5 million training steps for fairness. For each seed, we evaluated the final policy in 16 randomly initialized episodes lasting 1000 steps. We report the reward, which corresponds to the number of steps spent near the final goal of the task automaton, and the success rate ($S.R.$), which is the proportion of policy rollouts that completely satisfied the task specification for all 1000 steps. Because the robustness of rollouts for the final "Loop" task cannot be meaningfully evaluated over a finite trajectory [58], we instead compute the success rate by examining the proportion of rollouts that are never unsafe and successfully complete at least 1 full loop within the episode.

We find that in these environments, stably reaching subgoals with any non-goal-conditioned method is unreliable, especially when rewards are delayed based on the task structure. As a result, CRM-RS fails for almost all environments and task types. The LOF is able to more easily scale to handle multiple goals due to its hierarchical structure and achieves high robustness for most rollouts. However, it also doesn't typically obtain high rewards due to only learning to apply options for reaching subgoals, as opposed to remaining near the final goal once the task is satisfied. Furthermore, the tasks with safety constraints clearly demonstrate that rollout termination in CRM-RS and the reward shaping used by LOF is insufficient for reliably preventing the agent from entering obstacle regions. Our method obtained significantly more reward than the baselines, while remaining consistently safe, in all five tasks and environments.

## 5.2 Real-World Experiments

As a first step in determining our algorithm's applicability to controlling robots in the real world for LTL specified tasks, we trained a policy using ACQL for controlling a 6-DOF manipulator in a storage cabinet workspace. We trained this policy in the simulated environment pictured in Figure 2d for a complex navigation task involving 3 subgoals and 2 obstacles: $\Diamond(p_1 \wedge \circ(\Diamond(p_2 \wedge \circ(\Diamond(p_3))))) \wedge \Box(\neg(\text{in\_wall} \vee \text{in\_table}))$. The policy was trained over 6 actions corresponding to translating the end-effector in the 6 cardinal directions. The geometry of the simulated environment perfectly aligned with the real cabinet

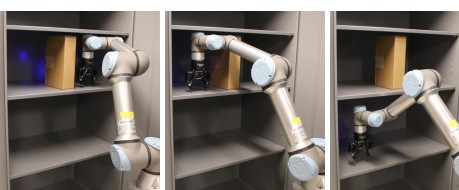

Figure 3: ACQL policies trained with safety constraints based a cabinet's geometry can be successfully deployed for a UR5e manipulator operating in the real cabinet environment.

workspace, so that policy actions based on the state of the simulation could be feasibly executed on the real robot. The resulting policy achieved a mean reward of $908.4$ and a $100\%$ success rate across 16 simulated rollouts, and we successfully deployed this same policy to the real UR5e robot arm and storage cabinet environment visualized in Figure 3. To support reproducibility, all relevant details for the task definition and environment setup are provided in our Supplementary Material. In future work, we intend to support a wider sim-to-real gap and train policies for LTL objectives in partially observable environments. Nonetheless, these initial results demonstrate that our algorithm can effectively leverage feedback from STL tasks in simulation to produce performant and safe policies for real robotic systems.

## 5.3 Ablative Analysis

We conducted an additional ablative analysis to justify two key components of our algorithm. The results are reported in Table 2. In the first ablation, we removed the use of HER from ACQL, requiring

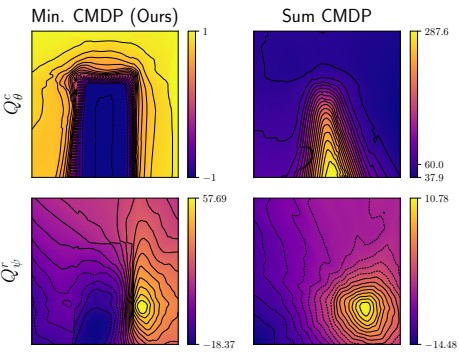

Figure 4: Contour plots for both $Q_\theta^c$ and $Q_\psi^r$ trained with ACQL with our CMDP formulation in (2) and the standard formulation in (1).

Table 2: Results for our algorithm when ablating HER and substituting our minimum-safety CMDP formulation with the standard CMDP formulation. We report mean and standard deviation of the reward and success rate collected over all environments, tasks (except the loop task), and seeds from the policy at the end of training in 16 evaluation episodes lasting 1000 steps. For the ablation affecting safety constraints, we only report mean performance in tasks that include safety constraints.

|  |  | Reward ↑ | $S.R.\% \uparrow$ |
|---|---|---|---|
| All Tasks | ACQL | **682.5 ± 232.1** | **91.5 ± 20.3** |
|  | No HER | 95.7 ± 220.1 | 12.9 ± 29.2 |
| Safety Tasks | ACQL | **545.8 ± 365.5** | **75.4 ± 39.1** |
|  | Sum CMDP | 8.0 ± 24.9 | 4.6 ± 11.0 |

the agent to learn exclusively from the sparse reward given upon reaching an accepting automaton state. This ablation isolates the effect of our subgoal-including product CMDP formulation that enables applying GCRL methods in the context of LTL tasks, and our results show that removing it leads to significantly degraded performance.

Our second ablation demonstrates the benefit of our minimum safety constraint (2) over the discounted sum-of-cost framework (1) for accurately learning the state-action safety function. We modified our CMDP to provide positive costs when there was a safety violation, trained $Q_\theta^c$ to predict discounted sums of costs, and constrained $\pi$ to choose actions below an upper limit on total cost $\mathcal{L}$. This limit was chosen based on the best performing value $\mathcal{L} \in \{0, 10, 40, 60\}$. We report the difference in performance over only the tasks that involved safety constraints and observe that our CMDP formulation is critical to the performance of our algorithm. We also show a visualization of the policy differences that result from this ablation in Figure 4. We observe that in our formulation (left), ACQL can more effectively learn the safety constraint, which substantially improves the quality of $Q_\psi^r$ (bottom). Note that the differing scale for $Q_\theta^c$ in our formulation (top left) results in a yellow safe region, while the standard formulation depicts low-cost, safe regions in blue.

# 6   Conclusion

**Limitations**   Although our method outperforms baseline methods in training policies to solve several fundamental types of LTL tasks, there remain a few limitations that we aim to address in future work. First, like many approaches leveraging product MDPs [13, 21, 32, 37], our method is restricted to logical tasks representable by DBAs. To overcome this, we plan to investigate more expressive types of automata for RL, such as Good-for-MDPs Non-Deterministic Büchi Automata [59]. Second, our current approach approximates the state-action value function without fully capturing task requirements beyond the current set of subgoals. We aim to explore more sophisticated goal-conditioned RL methods that can effectively condition a policy on future subgoals provided by our product CMDP. Finally, while our real-world deployment demonstrates the feasibility of applying our algorithm to robot control outside simulation, there are open challenges related to partial observability and sim-to-real mismatch in real-world learning with LTL objectives that we aim to address in future work.

We present a novel RL algorithm to tackle complex and high-dimensional LTL-specified tasks, enabled by a novel augmented product CMDP formulation that provides feedback for enforcing arbitrary safety constraints and subgoals for efficiently learning to achieve liveness constraints. By significantly outperforming comparable baselines for learning from LTL specifications, our approach demonstrates a promising trajectory toward learning in increasingly complex and realistic environments, without sacrificing generality in supporting the full expressiveness of LTL and other formal languages.

## Acknowledgments

This material is based upon work supported by the Air Force Office of Scientific Research under award number FA9550-24-1-0233. Any opinions, findings, and conclusions or recommendations expressed in this material are those of the author(s) and do not necessarily reflect the views of the United States Air Force.

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

# A Proof of Proposition 1

In what follows, we formally prove and restate Proposition 1, which shows that ACQL, under mild conditions, is guaranteed to return the optimal policy. Our proof is based on the proof of convergence for Q-learning using stochastic approximation theory in [60], extended with the theory of stochastic approximation under multiple timescales described in Chapter 6 of [53]. Refer also to [43, 55, 56] for similar analyses. The outline of the section is as follows:

1. Express ACQL as a stochastic approximation algorithm [53, 60].
2. Show that $Q^c$ and $Q^r$ converge to an optimal fixed point for any fixed safety discount factor $\gamma_c \in (0, 1)$.
3. Restate our Proposition 1 and prove it by showing that as $\gamma_c \to 1$, $Q^r$ and $Q^c$ converge to the optimal state-action value function $Q^{r*}$ and its corresponding optimal (undiscounted) state-action safety function $Q^{c*}$.

## A.1 Setup

**Assumption 1.** *The augmented CMDP $\mathcal{M}^A = (\mathcal{S}^A, \mathcal{A}, \mathcal{T}^A, d_0^A, r^A, c^A, \gamma, \mathcal{L})$ for ACQL is defined on a finite state space $\mathcal{S}^A$ and action space $\mathcal{A}$. For every state $s \in \mathcal{S}^A$ and action $a \in \mathcal{A}$, there is an associated bounded deterministic reward $r_{sa} = r^A(s, a)$ and bounded constraint feedback $c_{sa} = c^A(s, a)$ observed if action $a$ is applied at state $s$.*

Under Assumption 1, $Q^c$ and $Q^r$ are vectors in $\mathbb{R}^d$ where $d = |\mathcal{S}^A \times \mathcal{A}|$ is finite. The ACQL algorithm can be modeled as a distributed, asynchronous series of noisy updates to components of $Q^c$ and $Q^r$.

**Assumption 2.** *For each state-action pair $(s, a) \in \mathcal{S}^A \times \mathcal{A}$, there are an infinite number of updates applied to the components $Q^r_{s,a}$ and $Q^c_{s,a}$.*

The updates to these components are given by

$$Q^c_{s,a}(n+1) = Q^c_{s,a}(n) + a(n) \left[ \left( (1 - \gamma_c) c_{sa} + \gamma_c \min\{c_{sa}, \bar{Q}^c_{s',\pi(s')}(n)\} \right) - Q^c_{s,a}(n) \right] \text{ and}$$
(5)

$$Q^r_{s,a}(n+1) = Q^r_{s,a}(n) + b(n) \left[ (r_{sa} + \gamma \bar{Q}^r_{s',\pi(s')}(n)) - Q^r_{s,a}(n) \right],$$
(6)

where $s'$ is a randomly sampled next state following the state $s$ and action $a$. The elements of $\bar{Q}^c(n)$ and $\bar{Q}^r(n)$ are potentially taken from older iterations $Q^c_{s,a}(\nu_{s,a}(n))$ and $Q^r_{s,a}(\nu_{s,a}(n))$ where $\nu_{s,a}(n)$ is an integer satisfying $0 \leq \nu_{s,a}(n) \leq n$. Recall that the policy $\pi$ here is defined as $\arg\max_{a:\bar{Q}^c_{s,a} > \mathcal{L}} \bar{Q}^r_{s,a}$ in terms of $\bar{Q}^c$ and $\bar{Q}^r$. However, we assume that old information is eventually discarded as $n \to \infty$.

**Assumption 3.** *For all $(s, a)$, $\lim_{n \to \infty} \nu_{s,a}(n) = \infty$.*

Assumption 3 is necessary to prove the convergence of distributed asynchronous stochastic approximation algorithms using outdated values ($\bar{Q}^c$ and $\bar{Q}^r$) [60, 61]. Using $\bar{Q}^c$ is additionally necessary to define the operator in (8) such that a fixed policy $\pi$ can be used to prove the contraction property in Lemma 1.

We also update $\gamma_c$ infinitely often with

$$\gamma_c(n+1) = \gamma_c(n) + c(n) \left[ (1 - \gamma_c(n)) - \gamma_c(n) \right].$$
(7)

and assume that its updates are synchronized with the index $n$ for the updates to the components of $Q^c$ and $Q^r$.

**Assumption 4.** *The step sizes $a(n)$, $b(n)$, and $c(n)$ for the above updates satisfy*

$$\sum_{n=0}^{\infty} a(n) = \sum_{n=0}^{\infty} b(n) = \sum_{n=0}^{\infty} c(n) = \infty$$

$$\sum_{n=0}^{\infty} a(n)^2, \sum_{n=0}^{\infty} b(n)^2, \sum_{n=0}^{\infty} c(n)^2 < \infty,$$

*$b(n) \in o(a(n))$, and $c(n) \in o(b(n))$.*

Now, let $g : \mathbb{R}^d \times \mathbb{R}^d \times \mathbb{R} \to \mathbb{R}^d$ and $h : \mathbb{R}^d \times \mathbb{R}^d \times \mathbb{R} \to \mathbb{R}^d$ be operators defined for each component $(s, a)$ as

$$g_{s,a}(Q^r, Q^c, \gamma_c) = (1 - \gamma_c)c_{sa} + \gamma_c \, \mathbb{E}_{s'} \left[ \min\{c_{sa}, Q^c_{s',\pi(s')}\} \right] \tag{8}$$

$$h_{s,a}(Q^r, Q^c, \gamma_c) = r_{sa} + \gamma \, \mathbb{E}_{s'} \left[ Q^r_{s',\pi(s')} \right]. \tag{9}$$

Define a third mapping $f(Q^r, Q^c, \gamma_c) = 1 - \gamma_c$. Without loss of generality, we express all the operators as mappings from $\mathbb{R}^d \times \mathbb{R}^d \times \mathbb{R}$ to consider them as a coupled mapping from $\mathbb{R}^{2d+1}$ to $\mathbb{R}^{2d+1}$. Finally, define two martingale difference sequences

$$M^c_{s,a}(n+1) = \gamma_c \min\{c_{sa}, \bar{Q}^c_{s',\pi(s')}(n)\} - \gamma_c \, \mathbb{E}_{s'} \left[ \min\{c_{sa}, \bar{Q}^c_{s',\pi(s')}(n)\} \right] \text{ and} \tag{10}$$

$$M^r_{s,a}(n+1) = \gamma \bar{Q}^r_{s',\pi(s')}(n) - \gamma \, \mathbb{E}_{s'} \left[ \bar{Q}^r_{s',\pi(s')}(n) \right]. \tag{11}$$

The updates to $\gamma^c$ are deterministic. Using the above, we can express the three ACQL updates (5), (6), and (7) as

$$Q^c_{s,a}(n+1) = Q^c_{s,a}(n) + a(n) \left[ g_{s,a}(\bar{Q}^r(n), \bar{Q}^c(n), \gamma_c(n)) - Q^c_{s,a}(n) + M^c_{s,a}(n+1) \right], \tag{12}$$

$$Q^r_{s,a}(n+1) = Q^r_{s,a}(n) + b(n) \left[ h_{s,a}(\bar{Q}^r(n), \bar{Q}^c(n), \gamma_c(n)) - Q^r_{s,a}(n) + M^r_{s,a}(n+1) \right], \tag{13}$$

$$\gamma_c(n+1) = \gamma_c(n) + c(n) \left[ f(Q^r(n), Q^c(n), \gamma_c(n)) - \gamma_c(n) \right]. \tag{14}$$

Under the above assumptions and formulation, ACQL can now be analyzed as a distributed stochastic approximation algorithm under three timescales.

### A.2  Convergence of $Q^r$ and $Q^c$ under a fixed $\gamma_c$

Since the update to $\gamma_c$ happens much more slowly than the updates to $Q^r$ and $Q^c$—formally, the step size $c(n)$ for $\gamma_c$ shrinks faster than both $b(n)$ and $a(n)$—we can treat $\gamma_c$ as approximately fixed while analyzing the behavior of $Q^r$ and $Q^c$. This allows us to study the convergence of $Q^r$ and $Q^c$ assuming that $\gamma_c$ is a constant value in the interval $(0, 1)$.

**Lemma 1.** *The mapping $g^c = g(Q^r, \cdot, \gamma_c) : \mathbb{R}^d \to \mathbb{R}^d$, for some fixed $Q^r$, some fixed feasible policy $\pi$ (e.g., a policy based on $\bar{Q}^c$ and $\bar{Q}^r$ as in ACQL), and $\gamma_c \in (0, 1)$, is a contraction mapping.*

*Proof.*

$$
\begin{aligned}
|g^c(Q^c)_{s,a} - g^c(\hat{Q}^c)_{s,a}| &= |\gamma_c \, \mathbb{E}_{s'} \left[ \min\{c_{sa}, Q^c_{s',\pi(s')}\} \right] - \gamma_c \, \mathbb{E}_{s'} \left[ \min\{c_{sa}, \hat{Q}^c_{s',\pi(s')}\} \right] | \\
&= \gamma_c \, \mathbb{E}_{s'} \left[ |\min\{c_{sa}, Q^c_{s',\pi(s')}\} - \min\{c_{sa}, \hat{Q}^c_{s',\pi(s')}\}| \right] \\
&\le \gamma_c \, \mathbb{E}_{s'} \left[ |Q^c_{s',\pi(s')} - \hat{Q}^c_{s',\pi(s')}| \right] \quad (|\min\{a, b\} - \min\{a, c\}| \le |b - c|) \\
&\le \gamma_c \|Q^c - \hat{Q}^c\|_\infty
\end{aligned}
$$

Therefore, $\|g^c(Q^c) - g^c(\hat{Q}^c)\|_\infty \le \gamma_c \|Q^c - \hat{Q}^c\|_\infty$. $\qquad \square$

**Lemma 2.** *As $n \to \infty$, $Q^c(n)$ converges to a fixed point $\lambda_1(Q^r, \gamma_c)$ for some fixed $Q^r$ and $\gamma_c$.*

*Proof.* The convergence of $Q^c$ for a fixed $Q^r$ and $\gamma_c$ follows from Theorem 3 (convergence of distributed stochastic approximation algorithms for a contraction mapping) in [60]. Assumptions 1, 2, and 3 in [60] are satisfied due to our Assumptions 3, 4 and the definitions of $M^c_{s,a}$, $M^r_{s,a}$ in (10) and (11). Furthermore, the contraction property of $g^c$ (Lemma 1) is enough to satisfy Assumption 5 in [60]. Under these conditions, Theorem 3 in [60] holds true. $\qquad \square$

**Lemma 3.** *As $n \to \infty$, $Q^r(n)$ updated with (13) using a fixed $Q^c(n) = \lambda_1(Q^r(n), \gamma_c)$, such that there is a feasible action in every state, converges to the optimal value function $Q^{r^*}_{\gamma_c}$.*

*Proof.* The mapping $h^r = h(\cdot, Q^c, \gamma_c) : \mathbb{R}^d \to \mathbb{R}^d$ for a fixed $Q^c = \lambda(Q^r, \gamma_c)$ is a typical Bellman operator for a fixed policy using only actions from a constant non-empty subset of $\mathcal{A}$ for each state $s$. As a result, Theorem 4 (convergence of standard Q-learning) in [60] applies. □

**Lemma 4.** $Q^r(n)$ *and* $Q^c(n)$ *asymptotically approach* $Q_{\gamma_c}^{r^*}$ *and* $Q_{\gamma_c}^{c*} = \lambda_1(Q_{\gamma_c}^{r^*}, \gamma_c)$ *as* $n \to \infty$.

*Proof.* Lemmas 2 and 3 serve to satisfy Assumptions 1 and 2 in Chapter 6 of [53]. The boundedness of our rewards and constraint signals also result in a bounded $Q^r(n)$ and $Q^c(n)$, which satisfies Assumption 3 in Chapter 6 of [53]. The proof follows from Theorem 2 (convergence of two-timescale coupled stochastic approximation algorithms) in the same chapter. □

### A.3 Convergence of $(Q^r, Q^c)$ and $\gamma_c$

We can apply a similar two-timescale argument now using $\gamma_c$ on the slower timescale and $(Q^r(n), Q^c(n))$ on the faster timescale. The condition that the faster timescale converges to a fixed point $\lambda_2(\gamma_c)$ for a static $\gamma_c$ is shown in Lemma 4. For the condition that the slower timescale converges to a fixed point with $(Q^r(n), Q^c(n)) = \lambda_2(\gamma_c))$ is true trivially because $\gamma_c$ converges without depending on $(Q^r(n), Q^c(n))$ at all.

**Lemma 5.** $\gamma_c$ *converges to* 1 *as* $n \to \infty$.

*Proof.* The update in (14) is a discretization of the ODE $\dot{\gamma}_c(t) = 1 - \gamma_c(t)$. The solution to this ODE is $\gamma_c(t) = 1 - (1 - \gamma_c(0))e^{-t}$, which asymptotically approaches 1 as $t \to \infty$. □

Finally, similar to Proposition 1 in [16], we observe that $\lim_{\gamma_c \to 1} g_{s,a}^c(Q^c) = \min\{c_{sa}, \mathbb{E}_{s'} Q_{s',\pi(s')}^c\}$ yields a fixed point at $Q_{s,a}^{c*} = \mathbb{E}_{\tau \sim \pi}\left[\min_{t \in [0,\infty]} c_{sa_t}|s_0 = s, a_0 = a\right]$ matching the undiscounted minimum-safety constraint (Equation (2) in our main paper). Using this fact and another application of Theorem 2 in [53], we can prove our Proposition 2.

**Proposition 2.** *Let* $\mathcal{M}^{\mathcal{A}}$ *be an augmented CMDP with* $|\mathcal{S}^A| < \infty$, $|\mathcal{A}| < \infty$, *and* $\gamma \in [0, 1)$, *and let* $Q^c(n)$ *and* $Q^r(n)$ *be models for the state-action safety and value functions indexed by* $n$. *Assume they are updated using Robbins-Monro step sizes* $a(n)$ *and* $b(n)$, *respectively, with* $b(n) \in o(a(n))$ *according to (5) and (6). Assume that* $\gamma_c(n)$ *is also updated with step sizes* $c(n)$ *such that* $\gamma_c(n) \to 1$ *and* $c(n) \in o(b(n))$. *Then* $Q^c(n)$ *and* $Q^r(n)$ *converge to* $Q^{c*}$ *and* $Q^{r*}$ *almost surely as* $n \to \infty$.

*Proof.* By Theorem 2 from Chapter 6 of [53], whose conditions are satisfied by Lemmas 4 and 5, the coupled iterates $(\gamma_c(n), Q^r(n), Q^c(n))$ converge almost surely to a fixed point $(1, \lambda_2(1))$ with $\lambda_2(\gamma_c) = (Q_{\gamma_c}^{r*}, \lambda_1(Q_{\gamma_c}^{r*}, \gamma_c))$ as $n \to \infty$. As $\gamma_c \to 1$, $Q^c(n) = \lambda_1(Q_{\gamma_c}^{r*}, \gamma_c)$ also converges to $\mathbb{E}_{\tau \sim \pi}\left[\min_{t \in [0,\infty]} c_{sa_t}|s_0 = s, a_0 = a\right]$ with the policy $\pi$ determined by $\lim_{n \to \infty} \bar{Q}^r(n) = Q^{r*}$. Therefore, the algorithm attains the optimal state-action value function and the corresponding optimal (undiscounted) state-action safety function $Q^{c*}$. □

## B Additional ACQL Details and Pseudocode

**Automaton Analysis** To better illustrate the initial automaton analysis in ACQL (Line 2), consider the automaton in Figure 5. There is only one non-accepting sink-components of this automaton, and it is the component consisting of the single node $q_3$. For $q_0$, there is only one transition into this component via the edge labeled by $p_4$, so the safety for $q_0$ condition is $S(q_0) = \neg p_4$. For $q_1$ and $q_2$, there are no transitions to a non-accepting sink-component and so their safety conditions are $S(q_1) = S(q_2) = 1$, meaning that the safety condition is trivially satisfied at all times in those states. For completeness, we also consider the unsafe states themselves as having safety conditions equal to the conjunction of their negated incoming transitions. Therefore, $q_3$ also has the safety condition $S(q_3) = \neg p_4$.

Now we can obtain the liveness constraints, which are summarized in the liveness condition mapping $O : \mathcal{Q} \to \Phi$ (See Section 4.1 in our main paper). For $q_0$, the only remaining outgoing edge is the one labeled $\neg(p_1 \lor p_2) \land \neg p_4$. Since $S(q_0) = \neg p_4$, we can eliminate it from this transition predicate to obtain $O(q_0) = (p_1 \lor p_2)$. For $q_1$, one can simply obtain $O(q_1) = p_3$ from its only outgoing edge. For completeness, we also set $O(q_2) = p_3$ using the incoming edges for $q_2$ since it is an

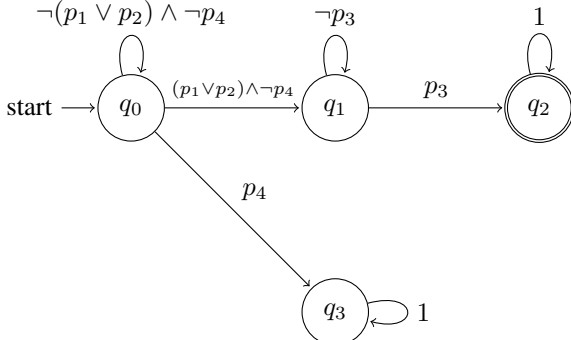

Figure 5: Automaton for the task "Reach goal $g_1$ or $g_2$ while never entering an unsafe-region $u_1$. Then reach $g_3$.", where achieving the goal $g_i$ corresponds to the atomic proposition $p_i$ and entering $u_1$ corresponds to $p_4$. The full LTL expression is $\neg p_4 \, \mathcal{U} \, ((p_1 \vee p_2) \wedge \circ \Diamond p_3)$. The proposition $p_4$ is only relevant to the task's safety constraint, and the propositions $p_1$, $p_2$, and $p_3$ are only relevant to the task's liveness constraints.

accepting state. This is purely to inform the agent of the goal associated with an accepting state once it has reached it. From the above values for $O$, the mapping $G$ can be defined by $G(q_0) = \{g_1, g_2\}$, $G(q_1) = \{g_3\}$ and $G(q_2) = \{g_3\}$.

**Subroutines** Algorithms 2, 3, and 4 present the pseudo-code for collecting trajectories in $\mathcal{M}^A$, relabeling them with achieved goals, and computing the safety discount factor $\gamma_c$.

The GetTrajectory function, in Algorithm 2, begins by sampling an initial state from the distribution $d_0^A$. We also randomly sample positions from the environment to associate with each subgoal proposition $p \in AP_{\text{subgoal}}$. This task randomization promotes collecting trajectories that explore a greater portion of the state space through a variety of subgoal sequences, and we found that this was necessary to stabilize training while using HER. We speculate that the greater variety of state and subgoal combinations is needed to train a robust subgoal-reaching policy. The agent proceeds to interact with the environment for a total of $T$ steps. In a loop indexed by $t$, actions are selected from an epsilon-greedy version of the input policy $\pi$ (Line 5). The reward, constraint feedback, and next state are observed (Lines 6-8) and stored in the trajectory $\tau$ (Line 9). After $T$ steps have executed, the trajectory is returned.

The Relabel function, in Algorithm 3, takes a batch of trajectories $\mathcal{B}_\tau$. For each trajectory in the batch, the function determines the final achieved state $g'$ and overwrites a single subgoal for every step in the entire trajectory with $g'$. It also overwrites the reward for each step with 1 for steps that were sufficiently close to $g'$ and 0 for steps that were not. We found that this relatively simple strategy, despite the fact that trajectories were collected with multiple-subgoals in-mind, was sufficient to train reliable goal-reaching policies.

The SafetyGammaScheduler function, in Algorithm 4, generates values for $\gamma_c$, starting from an initial value and gradually increasing toward 1.0 with exponential decay. To ensure accurate learning of state-action safety function models, it was necessary to cap $\gamma_c$ at a value slightly below 1.0.

## C  Model Architectures, Environment Implementation Details, and Hyper-Parameters

**Model Architecture and Policy** Our method trains two models: the state-action value function, $Q_\psi^r$, and the state-action safety function, $Q_\theta^c$. Both models employ twin neural networks and use the minimum of their predicted values to mitigate overestimation in value and safety estimation. For the ablation study using a state-action sum-of-costs function, we likewise use twin networks, but instead take the maximum of the two predictions to maintain a conservative (i.e., pessimistic) estimate.

To simplify handling multiple goals, $Q_\psi^r$ is defined using a goal-conditioned state-action value function model, $Q_\psi^{GC} : \mathcal{S} \times \mathcal{G} \times \mathcal{Q} \times \mathcal{A} \to \mathbb{R}$, parameterized by $\psi$. At runtime, $Q_\psi^{GC}$ is called for

---

**Algorithm 2** GetTrajectory

---

1: **function** GETTRAJECTORY($\mathcal{M}^A$, $\pi$)
2:     $s_0^A \sim d_0^A$, $g_i \sim P(\mathcal{G})$ for $p_i \in AP_{\text{subgoals}}$
3:     $t \leftarrow 0$, $\tau \leftarrow ()$
4:     **while** $t < T$ **do**
5:         $a_t \sim \pi_{\epsilon-\text{greedy}}(s_t^A)$
6:         $r_t \leftarrow r^A(s_t^A, a_t)$
7:         $c_t \leftarrow c^A(s_t^A, a_t)$
8:         $s_{t+1}^A \sim \mathcal{T}^A(s_t^A, a_t)$
9:         $\tau \leftarrow \tau \cup (s_t^A, r_t, c_t, a_t)$
10:        $t \leftarrow t + 1$
11:     **end while**
12:     **return** $\tau$
13: **end function**

---

**Algorithm 3** Relabel

---

1: **function** RELABEL($\mathcal{B}_r$)
2:     **for** $\tau \in \mathcal{B}_r$ **do**
3:         $g' \leftarrow$ the final goal state achieved in $\tau$.
4:         **for** $s_t^A, r_t \in \tau$ **do**
5:             Replace $g_1$ in $s_t^A$ with $g'$
6:             $r_t \leftarrow 1$ if $g'$ achieved in $s_t^A$ and 0 otherwise.
7:         **end for**
8:     **end for**
9:     **return** $\mathcal{B}_r$
10: **end function**

---

each subgoal $g \in g^+$. This approach exploits the fact that goal-conditioned value functions form a Boolean algebra under the $\min$ and $\max$ operators [21, 34]. For example, consider two subgoal propositions $p_1, p_2 \in AP_{\text{subgoal}}$. In the simplest case, when $O(q) = p_1$, we compute $Q_\psi^r(\langle s, g_1, q \rangle, a)$ as $Q_\psi^{GC}(s, g_1, q, a)$. When $O(q) = p_1 \wedge p_2$ (i.e., both subgoals must be achieved to progress), we compute the value as the minimum of $Q_\psi^{GC}(s, g_1, q, a)$ and $Q_\psi^{GC}(s, g_2, q, a)$. Conversely, when $O(q) = p_1 \vee p_2$ (i.e., achieving either subgoal suffices), the value is the maximum of $Q_\psi^{GC}(s, g_1, q, a)$ and $Q_\psi^{GC}(s, g_2, q, a)$. This pattern generalizes to arbitrarily complex Boolean formulae, allowing us to efficiently approximate $Q_\psi^r$ using a single goal-conditioned network.

We also observed that, because liveness constraints are separated into the goal input, conditioning behavior on the automaton state is only necessary when safety constraints differ between automaton states. As a result, we do not require a distinct "mode" for every automaton state $q \in \mathcal{Q}$, but only for each unique safety condition in the mapping $S$. To encode this, we train $Q_\psi^r$ using a multi-headed neural network, where each output head corresponds to a distinct safety condition. For example, in the automaton shown in Figure 5, the mapping $S$ assigns states to one of two safety conditions: $\neg p_4$ or 1. Accordingly, $Q_\psi^{GC}$ has two output heads, selected based on the currently active safety condition. In all ACQL experiments, $Q_\psi^{GC}$ shared a hidden layer of 256 neurons across all heads; each head then had an additional hidden layer of 256 neurons. All layers used ReLU activations, and the final output layer had $|\mathcal{A}|$ neurons with no activation function.

The model $Q_\theta^c$ follows the same architecture as $Q_\psi^r$, with a goal-conditioned, multi-headed neural network, but differs in two key respects. First, for disjunctive goal conditions ($p_1 \vee p_2$), the output is defined as the minimum of $Q_\theta^{GC}(s, g_1, q, a)$ and $Q_\theta^{GC}(s, g_2, q, a)$, to ensure conservative safety by taking the worst-case estimate across disjunctive paths. Second, $Q_\theta^{GC}$ uses a different network architecture: it has two shared hidden layers of 64 neurons each, and each output head includes two additional hidden layers with 64 and 32 neurons, respectively. All layers use ReLU activations. The final output layer consists of $|\mathcal{A}|$ neurons with a $\tanh$ activation function.

The policy was implemented in terms of these two value functions according to the constrained maximization $\pi(s) = \arg\max_{a:Q_\theta^c(s,a)>\mathcal{L}} Q_\psi^r(s, a)$. In the case that no action was deemed feasible by $Q_\theta^c$, the safest action $\max_a Q^c(s, a)$ was chosen. The exploration policy $\pi_{\epsilon-\text{greedy}}$ would behave

---
**Algorithm 4** SafetyGammaScheduler
---
1: **function** SAFETYGAMMASCHEDULER($j$)
2:     x ← $j$ ÷ update_period
3:     y ← $1.0 - (1.0 - \text{init\_value}) \cdot \text{decay\_rate}^{\text{x}}$
4:     **return** $\begin{cases} \text{max\_value} & \text{if y} \geq \text{max\_value} \\ \text{y} & \text{o.w.} \end{cases}$
5: **end function**
---

as above with probability $1 - \epsilon$, and with probability $\epsilon$ select a random action without considering $Q^r$ or $Q^c$.

**Environment Implementation**    For our simulated experiment environments, we used the Brax physics simulator [57] and assets provided in JaxGCRL [62] for the PointMass, Quadcopter, and Ant environments. Acting in these environments was facilitated by a set of discrete actions corresponding to movement in the cardinal directions. Specifically, the actions in the PointMass and Quadcopter environments output a constant low-level action to accelerate in one of the four or six available directions for 5 consecutive steps. The actions for our UR5e environment, which we used to train our real-world-deployed policies, similarly moved the robot's end effector in the 6 cardinal directions for a single time step. The actions in the AntMaze environment were policies trained separately using Proximal Policy Optimization (PPO) [7] for 50000000 environment interactions with the objective of maximizing velocity in each of the four cardinal directions and would run for 4 consecutive steps when executed. All further details regarding environment geometry and task definitions for our simulated and real-world experiments are included in our Code Repository[1].

**Hyper-Parameters**    Table 3 reports the hyperparameter values most commonly used in our experiments, including hyperparameters for the safety gamma ($\gamma_c$) scheduler described in Appendix B. For a complete account of hyperparameters, as well as ACQL, baseline, and environment implementation details, refer to our Code Repository[2][1].

Table 3: Hyperparameter values used for experiments in Tables 1 and 2 in our main paper

| Hyperparameter Name | Value |
|---|---|
| Episode length ($T$) | 1000 |
| Discount factor ($\gamma$) | 0.99 |
| Learning rate ($\alpha$) | $1 \cdot 10^{-4}$ |
| $\epsilon$-greedy factor ($\epsilon$) | 0.1 |
| Safety limit ($\mathcal{L}$) | 0.0 |
| Safety Gamma Init Value | 0.80 |
| Safety Gamma Update Period | 250,000 |
| Safety Gamma Decay Rate | 0.15 |
| Safety Gamma Max. Value | 0.98 |
| Target parameter interpolation factor ($\lambda$) | 0.005 |

**Compute Resource Requirements**    All experiments were conducted on a single NVIDIA RTX 3090 GPU (24 GB VRAM), using a local workstation equipped with an 12th Gen Intel i7-12700F CPU, 32 GB RAM. No cloud services or compute clusters were used. Each individual experimental run required approximately 30 minutes of compute time on the GPU. The full experiment grid consists of 225 runs for the comparative analysis and 90 runs for the ablations, amounting to approximately 315 GPU-hours. Minor additional compute was used for initial hyperparameter tuning and development.

# D    Expanded Experimental Results

Figures 6 and 7 show the average reward and success rate throughout training for the baseline comparison experiments summarized in Table 1. The LOF baseline cannot be depicted on these

---
[1]https://github.com/Tass0sm/acql

plots as it does not learn a single policy in the same MDP as the other methods, and instead learns a policy that chooses subgoal-specific options for an abstracted state space $\mathcal{Q} \times \mathcal{S}_g$ constructed from the automaton states $\mathcal{Q}$ and the finite set of states $\mathcal{S}_g \subset \mathcal{S}$ corresponding to task subgoals. Figures 8 and 9 show the average reward and success rate throughout training for the experiments summarized in Table 2 in our main paper. The differences in amount of training steps depicted by the figures is due to the different design and training pipelines that the two algorithms observe. ACQL collects complete trajectories to store in the Replay Buffer and CRM-RS just collects individual transitions. We want to highlight that our algorithm converges earlier during the training and this difference does not play a significant role in the performance gap reported in Table 1.

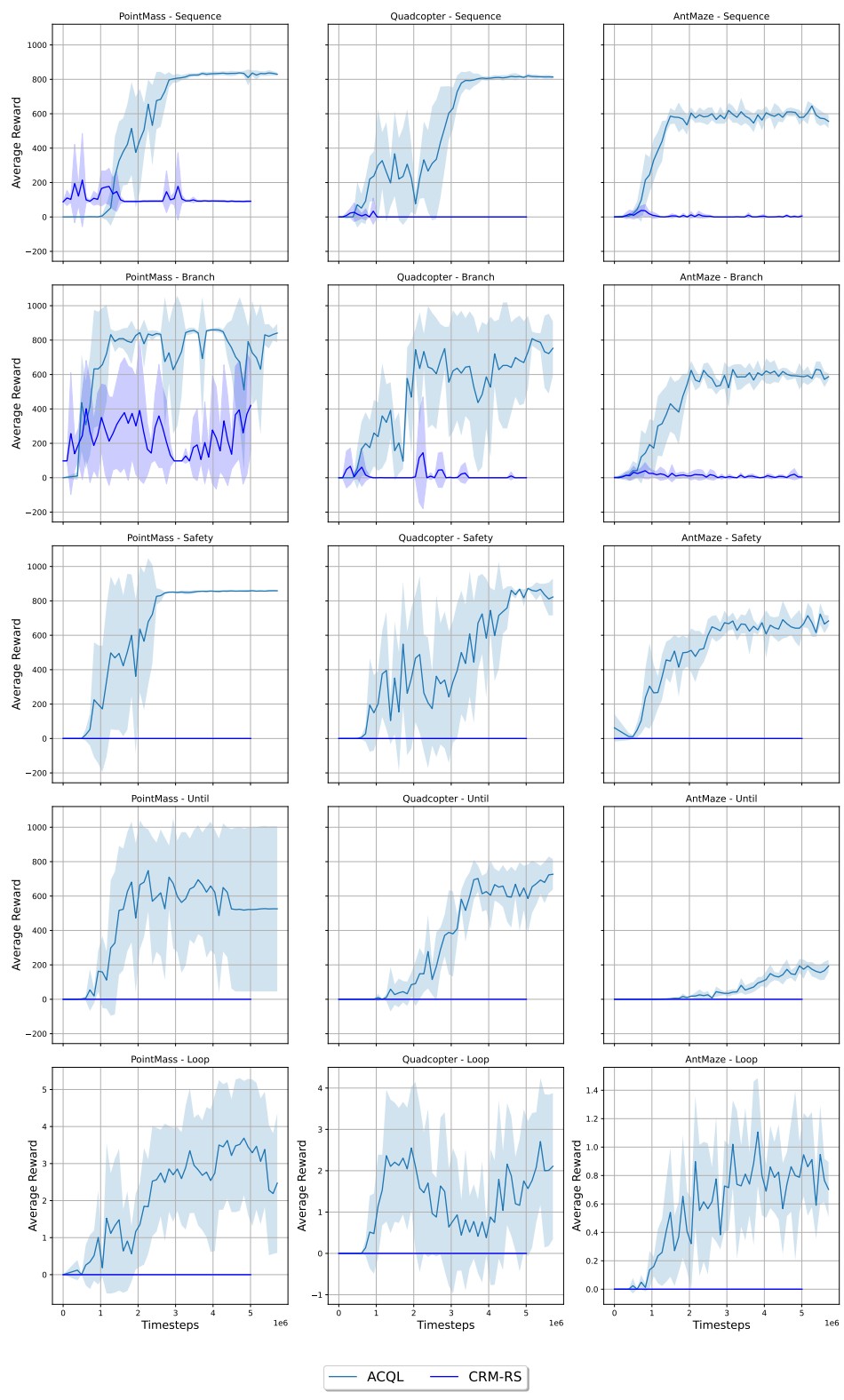

Figure 6: Average and one standard deviation of episode reward throughout training for the five runs per method that are summarized in Table 1 in our main paper.

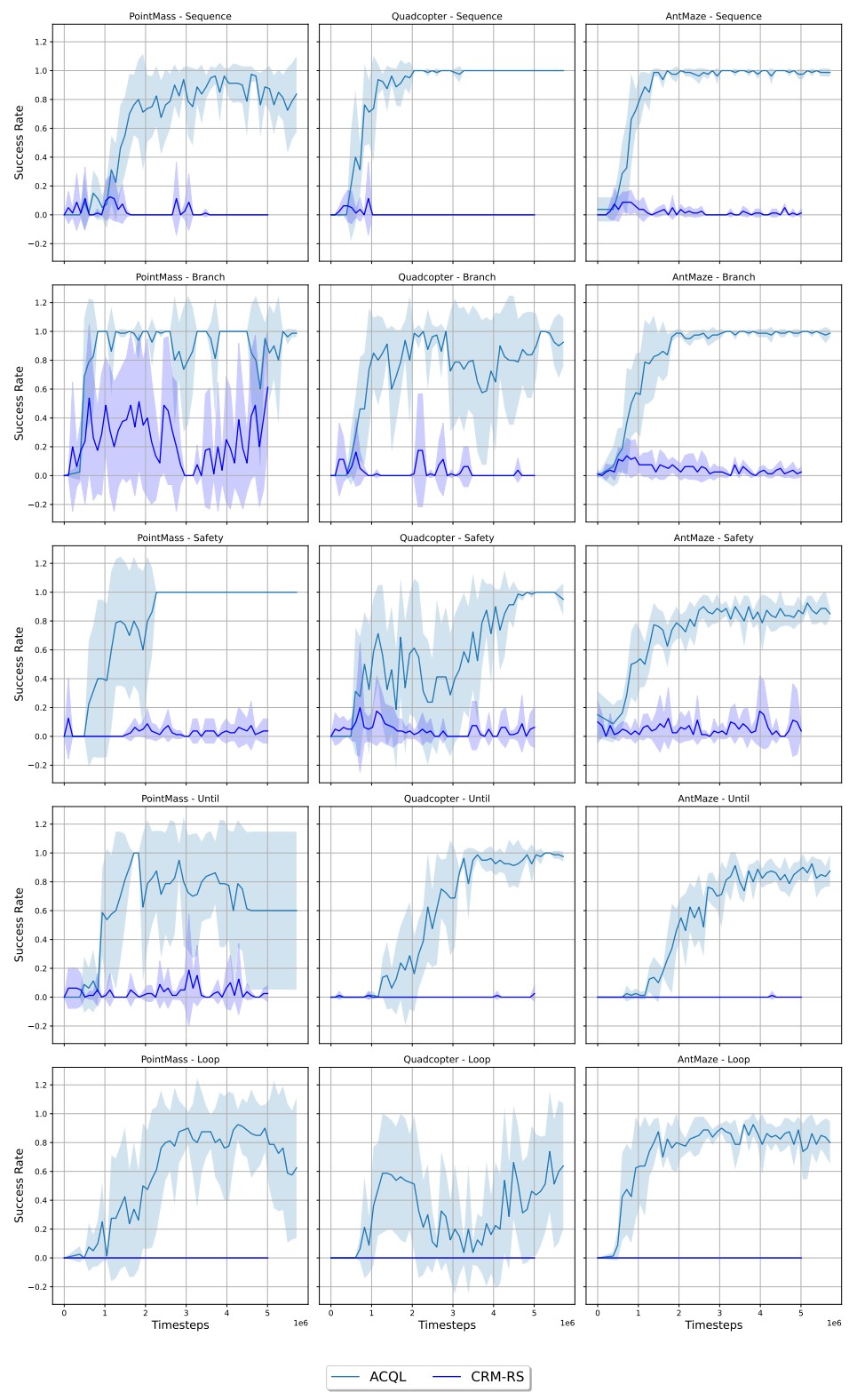

Figure 7: Average and one standard deviation of episode success rate throughout training for the five runs per method that are summarized in Table 1 in our main paper.

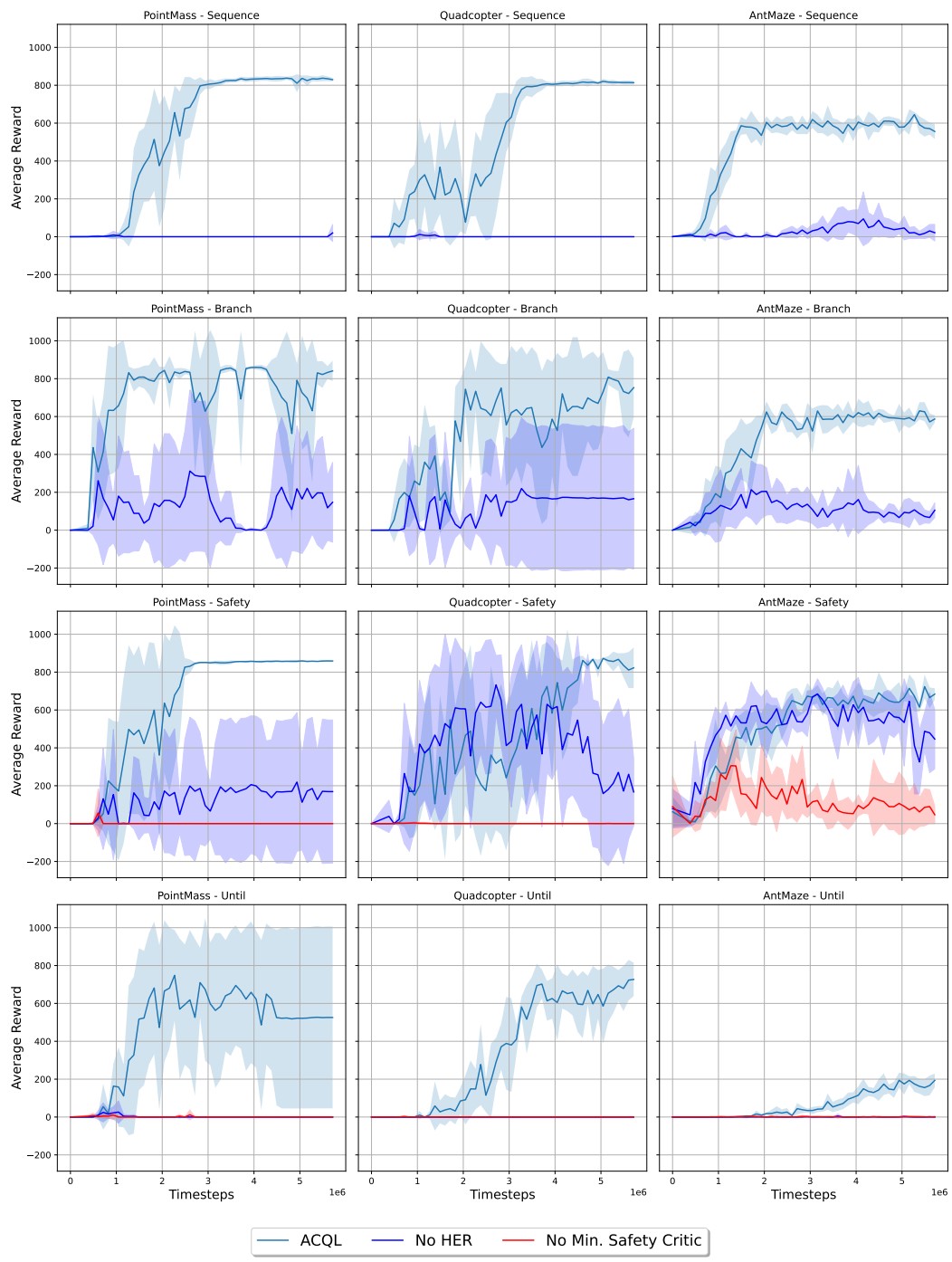

Figure 8: Average and one standard deviation of episode reward throughout training for the five runs per ablation group that are summarized in Table 2 in our main paper.

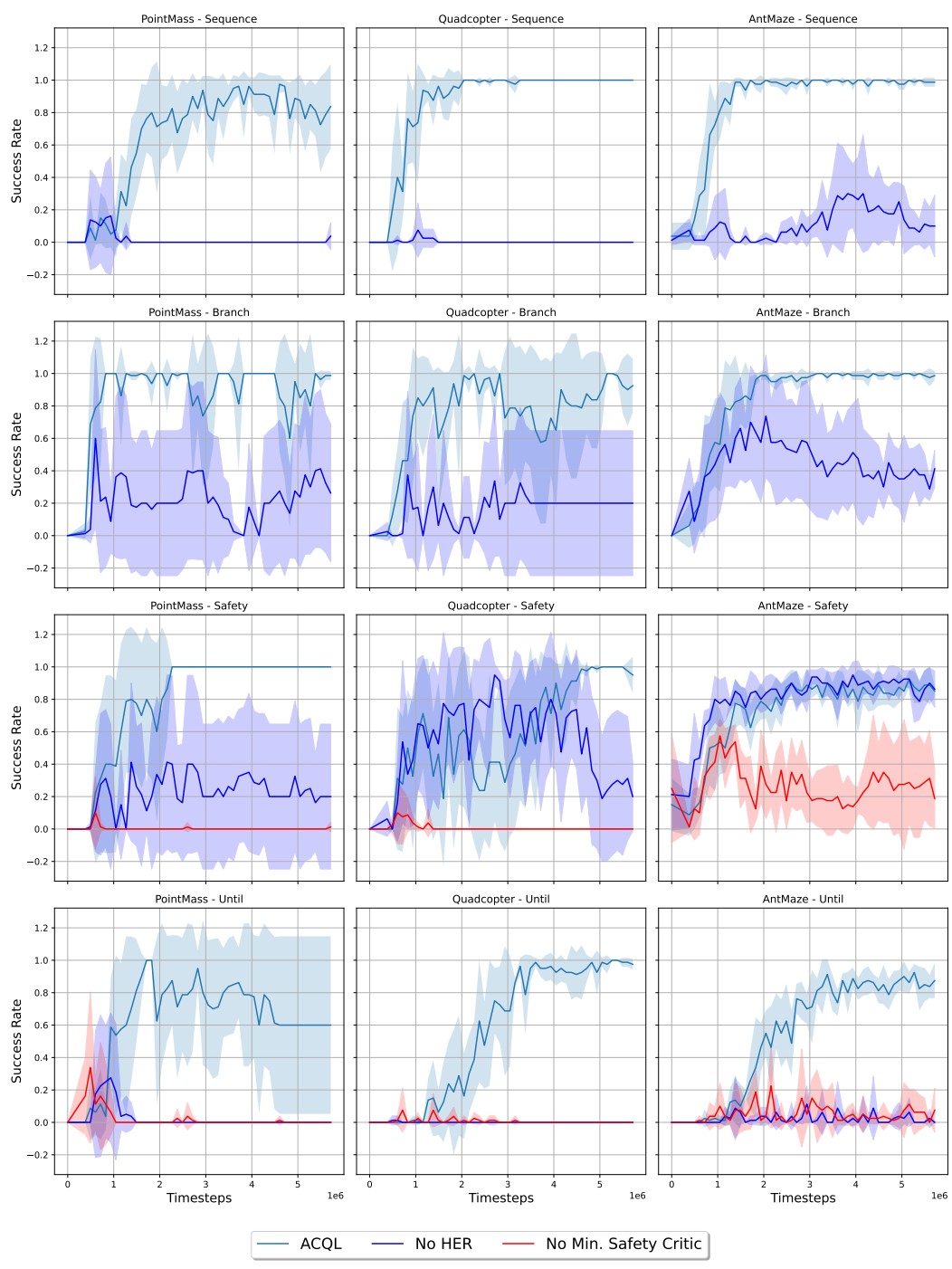

Figure 9: Average and one standard deviation of episode success rate throughout training for the five runs per ablation group that are summarized in Table 2 in our main paper.

