# OpenReview forum: "Automaton Constrained Q-Learning"
_NeurIPS.cc/2025/Conference — NeurIPS 2025 poster_

### Official Review · Reviewer_v818 · 2025-06-24

**Clarity:** 3
**Significance:** 2
**Originality:** 2
**Rating:** 4
**Confidence:** 3

**Summary:**

The paper introduces Automaton Constrained Q-Learning, an algorithm for learning Reinforcement Learning (RL) policies for objectives given as Linear Temporal Logic (LTL) formulas. The method combines the use of LTL objectives in RL and safe RL techniques by rewarding based on the acceptance of the induced automaton of the given LTL formula and enforcing inferred safety constraints during training. Specifically, for the given LTL task, its safety and liveness components are extracted and used as reward and cost, respectively, in a product CMDP formulation while applying constraints based on the minimum over all future values of the cost, instead of a sum-based formulation. Then, separate Q-functions are learned simultaneously for reward and cost. Optimal actions are chosen w.r.t. the reward Q-values among the safe actions given by thresholding the cost Q-values. The authors present a proof of convergence for their algorithm as well as experiment results both in simulation and real-world settings.

**Questions:**

1. What is the main reason behind the sim2real success? Is it thanks to the proposed approach? Or was it an engineering effort? Have you tried any of the baselines in this setting? If so, what was their success rate?
2. How do you pass the subgoal lists to the networks? Is it possible to use LTL formula embeddings [5] or automata embeddings [6, 7] instead of subgoal lists? If so, what is the advantage of using subgoal lists over such embeddings?
3. Not algorithmically, but in terms of what is achieved, what are the differences between DeepLTL [1] and the proposed approach? Does the proposed approach provide anything that DeepLTL doesn't?

**Ethical Concerns:**

["NO or VERY MINOR ethics concerns only"]

**Final Justification:**

The authors have adequately addressed my concerns in their rebuttal; therefore, I am increasing my rating by one point (3 --> 4), assuming the final version of the paper incorporates the clarifications provided in the rebuttal and the discussion of the highlighted related work.

**Limitations:**

yes

**Paper Formatting Concerns:**

No major formatting issues.

**Quality:**

2

**Strengths And Weaknesses:**

Strengths:
- The paper presents a novel algorithm combining LTL objectives in RL and safe RL techniques in a non-trivial way. Developing learning algorithms tailored for the use of formal specifications in RL is a relevant contribution.
- The proof of convergence for the algorithm seems sound, but I went through the proof only once so definitely it needs to be checked again in more detail. That being said, from the theoretical formulation of the method, the convergence result is intuitive and as expected. I even wonder if there could be a shorter and easier-to-read proof for the same result.
- The real-world experiment performed on a 6-DOF robotic arm is one of the most interesting aspects of the paper, especially given that the training was done in a simulation environment and that the physical robot had a 100% success rate in real-world tests. I wonder if this sim2real success was an engineering effort or something enabled by the choice of minimum safety constraint or any other design choice made by the authors.


Weaknesses:
- In my opinion, the main weakness of the paper is that the time for this paper might have already passed. Specifically, there have been significant efforts and successes in the literature for using LTL objectives in RL over the last decade, some of which are cited in the paper as well. So, even though the proposed algorithm is a novel contribution to the best of my knowledge, its impact is debatable. Moreover, I think with a little bit more effort, the approach can be extended to multi-task settings, i.e., the same policy performing a large class of LTL tasks, or even, probably with more effort, to multi-agent ones.
- Some important related work is missing [1, 2, 3, 4, 5, 6, 7]. DeepLTL [1] is a relevant baseline, and the authors should definitely consider comparing against this work both in simulation and real-world settings. A discussion of the difference between the proposed approach and [2, 3] is needed. Using HER in a multi-task RL context was proposed in [4]. Also, I am curious if LTL formula embeddings [5] and automata embeddings [6, 7] can be used here instead of the subgoal lists in the product CMDP definition, as these embeddings can be a compact way to substitute the same information (and probably even more) provided by subgoal lists. Overall, it is hard to properly position the paper in the literature without these discussions and a comparison with DeepLTL.
- There are some issues with the formalism. In line 133, it is said that the alphabet of the automaton is the MDP states, which is fine in discrete state spaces, but if the state space is continuous, then the alphabet is uncountably infinite. So, I would recommend explicitly defining a labeling function as is usually done in the literature. In line 137, it is said that there are propositional formulas on the transitions that hold for all symbols in the alphabet, then does that mean any symbol would trigger any transition on the automaton? It is unclear what is meant here. Also, such a formulation, i.e., using propositional formulas on transitions, is almost describing a symbolic automaton rather than a Büchi automaton. In line 164, to define G^+, G^i is used but it is not clear what G^i is. It is later informally clarified in the paper but that initial part is confusing. In line 196, it is said that subgoal lists facilitate the use of HER, it is not clear why this is the case. HER can be easily applied in a standard product MDP definition (without subgoal lists) as well. Also, including Algorithm 1 (or a shorter version of it) in the main body of the paper might be helpful to the reader.

[1] Jackermeier, Mathias, and Alessandro Abate. "Deepltl: Learning to efficiently satisfy complex ltl specifications for multi-task rl." The Thirteenth International Conference on Learning Representations. 2025.

[2] Shah, Ameesh, et al. "LTL-Constrained Policy Optimization with Cycle Experience Replay." arXiv preprint arXiv:2404.11578 (2024).

[3] Xu, Duo, and Faramarz Fekri. "Generalization of temporal logic tasks via future dependent options." Machine Learning 113.10 (2024): 7509-7540.

[4] Yalcinkaya, Beyazit, et al. "Automata conditioned reinforcement learning with experience replay." NeurIPS 2023 Workshop on Goal-Conditioned Reinforcement Learning. 2023.

[5] Vaezipoor, Pashootan, et al. "Ltl2action: Generalizing ltl instructions for multi-task rl." International Conference on Machine Learning. PMLR, 2021.

[6] Yalcinkaya, Beyazit, et al. "Provably Correct Automata Embeddings for Optimal Automata-Conditioned Reinforcement Learning." arXiv preprint arXiv:2503.05042 (2025).

[7] Yalcinkaya, Beyazit, et al. "Compositional automata embeddings for goal-conditioned reinforcement learning." Advances in Neural Information Processing Systems 37 (2024): 72933-72963.

---

> ### Author Rebuttal · Authors · 2025-07-30
>
> Thank you for your insightful review and for taking the time to engage critically with our work.
>
> We have addressed each of your points in the responses below. If our clarifications resolve your concerns, we would be grateful if you would consider raising your score. Thanks again for your feedback, which has helped improve our work.
>
> ## Weaknesses
>
> > Even though the proposed algorithm is a novel contribution to the best of my knowledge, its impact is debatable.
>
> > The approach can be extended to multi-task or multi-agent settings.
>
>
> We want to emphasize that the contributions of our work lie not only in the technical components of the algorithm itself, such as the integration of HER with a goal-providing product MDP and Temporal Logic-derived minimum-safety constraints, but also in the accompanying theoretical analysis and empirical performance..
>
> While the use of temporal logic in RL has seen progress over the past decade, we believe that key challenges still remain, particularly when scaling to continuous, sparse-reward, and safety-critical environments. Our method directly targets this space and advances the state of the art in integrating temporal logic with Deep RL.
>
> We agree that extending this approach to multi-task and multi-agent settings would be a valuable next step. These directions pose non-trivial challenges and, in our view, our work provides a strong foundation upon which such work can be built.
>
> > Some important related work is missing [1, 2, 3, 4, 5, 6, 7].
>
> Thank you for bringing these works to our attention. We agree that they are highly relevant and we will include them in the final version of the paper.
>
> > DeepLTL [1] is a relevant baseline, and the authors should definitely consider comparing against this work.
>
> > In terms of what is achieved, what are the differences between DeepLTL [1] and the proposed approach?
>
>
> DeepLTL focuses on a different setting. It is designed for the multi-task paradigm, where the objective is to generalize across a distribution of LTL specifications using learned task embeddings. In contrast, our method targets single-task learning in high-dimensional continuous environments, with an explicit focus on safety.
>
> Because of this difference, applying DeepLTL in our setting would require disabling several of its core mechanisms, such as multi-task generalization and curriculum-based training. For this reason, we believe a direct comparison would not be meaningful.
>
> Additionally, DeepLTL places less emphasis on the RL backbone and on safety. It uses standard PPO within classical MDPs, whereas we adopt a CMDP formulation consistent with the safe RL literature, and introduce a product-CMDP construction along with a theoretical analysis.
>
> From examining the experimental domains featured in both works, we believe a key distinction is our method’s ability to learn effective policies in environments with complex robot dynamics and explicit safety requirements. In particular, our use of minimum-safety constraints instead of rollout termination results in greater robustness and safety guarantees.
>
> We view these two approaches as complementary. DeepLTL advances generalization across tasks, while our method focuses on safely and effectively satisfying a single temporal logic specification in complex continuous domains. Exploring how to combine the strengths of both approaches is a promising direction for future research.
>
> > A discussion of the difference between the proposed approach and [2, 3] is needed. Using HER in a multi-task RL context was proposed in [4].
>
> Thank you for pointing this out. We will include these references and discuss them in the Related Work section of the final version of the paper.
>
> Specifically, the work in [2] is also orthogonal to our work and provides a technique for providing additional rewards based on the agent’s partial completion of accepting cycles in the task automaton.
>
> The work in [3] is also a multi-task RL approach that, similar to the LOF [8], trains an option for several goals and also conditions each option on a representation of the possible sequence of subsequent goals. [4] also is an interesting approach that uses HER directly with goals based on embeddings of task automata, which we believe is distinct from our method’s use of HER for densifying rewards in a single-task setting.
>
> > LTL formula embeddings [5] and automata embeddings [6, 7] can be used here instead of the subgoal lists in the product CMDP definition
>
> > What is the advantage of using subgoal lists over formula embeddings [5] or automata embeddings [6, 7]?
>
>
> Thank you for this question. We agree that using LTL formula embeddings or automata embeddings is an exciting direction, particularly in the context of multi-task learning.
>
> In the single-task setting, we find it more effective to rely on explicit subgoal lists. Compressing an entire temporal logic objective into a single embedding introduces a layer of abstraction that may obscure the precise semantics of the task, making learning harder and less interpretable. In contrast, subgoal lists offer a direct representation of the automaton's state and progression, which provides clear task guidance at each step.
>
> Understanding how to combine the interpretability and task specificity of subgoal-based representations with the scalability and generalization potential of embedding-based approaches is an exciting direction for future work.
>
> > In line 133, it is said that the alphabet of the automaton is the MDP states. if the state space is continuous, then the alphabet is uncountably infinite.
>
> We agree that defining a labeling function $L : \mathcal{S} \to 2^{AP}$ and the automaton in terms of the alphabet $\Sigma = 2^{AP}$, as opposed to directly defining the automaton transition function as $\delta: Q \times \Sigma \to Q$ with $\Sigma = \mathcal{S}$, will improve clarity and our consistency with the literature. We are considering accordingly modifying our definitions in the final version of the paper.
>
> > In line 137, It is unclear what is meant by propositional formulas on the transitions that hold for all symbols in the alphabet.
>
> We introduce this definition of transition predicates, inspired by the terminology on page 3 of [9], to more easily discuss the implications of the automaton transition function $\delta$. These predicates still only are true for a subset of symbols, so it is not the case that any symbol can trigger any transition, and we believe this terminology is still compatible with Büchi automata.
>
> > In line 164, it is not clear what G^i is.
>
> The $\mathcal{G}^i$ refers to the i-fold cartesian product of $\mathcal{G}$ with itself, and we agree this should be made more clear. We intend to resolve this in the final version for the paper.
>
> > It is not clear why subgoal lists facilitate the use of HER. Also, including Algorithm 1 in the main body of the paper might be helpful to the reader.
>
> An obstacle to applying HER to a standard product MDP is that the standard product MDP defines no notion of goal that is compatible with the temporal logic task. HER would require such a goal in order to perform relabeling for the episode, but adding goal-conditioning while still supporting the task specified by the reward machine is not trivial. Other methods require introducing another representation of goals in order to enable goal-conditioning, such as the reach-avoid sequences in DeepLTL or automata embeddings in [4]. Our work instead introduces the subgoal lists to enable goal conditioning and HER for introducing additional rewards. Without this augmentation, HER cannot be easily applied in a standard product MDP.
>
> We will do our best to include Algorithm 1 in the final version of the paper.
>
> [8] Araki, Brandon, et al. "The logical options framework." International Conference on Machine Learning. PMLR, 2021.
>
> [9] Alpern, Bowen, and Fred B. Schneider. "Recognizing safety and liveness." Distributed computing 2.3 (1987): 117-126.
>
> ## Questions
>
> > What is the main reason behind the sim2real success?
>
> The sim-to-real success can be attributed primarily to the close alignment between the simulation and the real-world environment. The simulated setup used a state space and action space that were not highly affected by sensor noise, matched the real robot’s dynamics, and accurately modeled physical constraints such as the shelf. Training and evaluation were carried out entirely in this simulator.
>
> This type of sim-to-real transfer is common in policy learning when the simulation is carefully constructed to reflect the real-world system.
>
> > Have you tried any of the baselines in this setting? If so, what was their success rate?
>
> Based on initial evaluation in the simulator, we were able to determine if trained policies were safe to deploy or not. The baselines did not perform well in the simulated environment, and therefore we did not believe it was appropriate to deploy them in the real world.
>
> > How do you pass the subgoal lists to the networks?
>
> The subgoal list is processed by the networks according to how they appear in the boolean propositional statement $O(q)$ (defined in line 162) where $q$ is the agent’s current automaton state. Each goal is passed to a single-goal-conditioned value function module, and the output values for each goal are then combined according to min and max operations based on the “and” and “or” operators in $O(q)$ until a final value is obtained for the goal list. For more details, refer to the Appendix C Paragraph 1.

---

> > ### Comment · Reviewer_v818 · 2025-08-05
> >
> > I thank the authors for addressing my concerns and strongly recommend incorporating the clarifications from the rebuttal, along with the discussion of the highlighted related work, into the final version of the paper.

---

> > > ### Author Response · Authors · 2025-08-05
> > > **Thank you and confirmation for adding clarifications and related work**
> > >
> > > Thank you again for your time and effort in reviewing our paper. We want to reaffirm our commitment to incorporating the clarifications and the related work your review has highlighted into the final version of the paper. If every concern has been addressed, we will greatly appreciate your updated score. We would also be happy to answer any other questions and to discuss any other points of the paper. Thank you!

---

### Official Review · Reviewer_Lw5u · 2025-06-24

**Clarity:** 3
**Significance:** 3
**Originality:** 3
**Rating:** 5
**Confidence:** 4

**Summary:**

The paper introduces ACQL, a reinforcement learning algorithm that integrates goal-conditioned RL with tasks representable as deterministic Buchi automata (this corresponds to a sub-fragment of LTL properties). ACQL addresses scenarios where an agent must achieve temporally ordered sub-goals while respecting time-varying (dynamic) safety constraints – a setting that classical RL struggles with. The key idea is to translate an LTL task specification into a corresponding deterministic Büchi automaton (DBA) and augment the standard Markov Decision Process (MDP) into a product Constrained MDP (CMDP). In this augmented CMDP, each state is a tuple combining the original environment state, the automaton state, and a list of pending sub-goals extracted from the automaton. This construction yields a Markovian representation of the temporal task: the automaton state tracks progress toward task completion, and the sub-goal list encodes the next objectives required by the LTL formula. By encoding automaton-derived sub-goals into the state, ACQL can leverage goal-conditioned techniques (notably Hindsight Experience Replay, HER) to densify sparse rewards. To enforce dynamic safety constraints, the authors propose a novel constraint formulation: ACQL imposes a constraint on the minimum safety value along a trajectory, rather than the usual cumulative cost constraint.

**Questions:**

1.  Could the authors compare ACQL conceptually to approaches that use shielding or model-checking at runtime (to prevent unsafe actions), or to other LTL-guided RL methods (for instance, those by Hasanbeig et al. on deep LTL RL)?

2. Conceptually it makes sense $\gamma_c \to 1$ since you're taking a minimum over the infinite time horizon, in practice does this introduce instability issues? How sensitive is ACQL to the scheduling of the safety discount factor $\gamma_c$ and the learning rates on the two timescales? I am not familiar with [16] but could you explain how the contraction property can still be shown as $\gamma_c \to 1$? Further, did you have to tune the rate at which $\gamma_c$ approaches 1, or did a simple schedule (e.g., linear annealing) work out-of-the-box?

3. Does the training process itself ever cause safety violations in the environment? For example, in the simulation, if the agent takes an unsafe action, is the episode terminated or is it allowed to continue with a cost? In real-robot experiments, how were safety violations handled to protect the hardware?

4. How do the authors envision extending their approach to more general STL/LTL classes or more complex automata?

5. How can ACQL be adapted to partially observable settings?

**Ethical Concerns:**

["NO or VERY MINOR ethics concerns only"]

**Limitations:**

The authors explicitly discuss the limitations of their work in the conclusion section. They acknowledge three main limitations: expressiveness of tasks, value function approximation, and real-world partial observability.

### Societal Impact

The authors state that their research has negligible foreseeable societal impact regarding common concerns like human safety, security, discrimination, I tend to agree with their sentiment.

**Quality:**

4

**Strengths And Weaknesses:**

### Strengths

- This work offers a fresh perspective on the integration of formal methods with deep RL, by combining deterministic automata (from LTL/STL) with goal conditioned RL.

- The idea of an augmented product CMDP with sub-goals is novel and addresses the reward sparsity problem in a principled way, in addition, the minimum-safety constraint is an innovative adaptation of reachability analysis to the RL setting, and much more appropriate for safe RL than the usual expected cumulative cost constraint.

- The paper's methodology is solid and well-founded. The augmented CMDP construction is logically sound – by incorporating the automaton state, the LTL-defined reward becomes Markovian, which justifies applying Q-learning, in addition, the use of sub-goal lists for HER has empirical grounding in goal-conditioned value function theory.

- Importantly, the authors include a convergence proof (under mild conditions) for ACQL’s learning algorithm, showing that the two-timescale Q-learning scheme for reward and safety values will converge to the optimal solution with probability 1.

- Clarity is generally ok.

- The experimental evaluation is reasonable, particularly the extension to the real-world robot is interesting.


### Weaknesses

-  A notable limitation is that ACQL (like many automaton-based RL methods) is currently restricted to tasks that can be captured by a Deterministic Büchi Automaton (DBA), i.e., belong to the recurrence class.

- The authors choose not to compare to standard Safe RL methods, this is justified by the statement "Safe RL methods typically address stationary safety constraints", it is not clear what they mean by stationary safety constraints (see Questions), do they mean temporally extended safety constraints, I am assuming the safety constraints (encoded in the LTL formula) don't change during learning/execution thus they are stationary, in theory one could apply CMDP methods to the product CMDP, I imagine they wouldn't do well as they can't effectively deal with reward sparsity. It would be nice if the authors could address this caveat.

- Most of the gains seem to come from integration of HER (for more efficient learning of sub goals) and CMDP/safe RL (for biasing the class of policies), generally separating reward and safety is more efficient because it divides and conquers the different objectives, the authors claim that the baselines "RMs and the LOF cannot be easily enhanced with high-performing GCRL and Safe RL techniques without adopting substantial modifications to their learning algorithms and product MDP formulation" its not clear to me why this is the case, I could imagine extending RMs with HER and some Lagrange relaxation of CMDP for the Bellman update step, this might be non-trivial but I imagine the performance would be much closer to ACQL. While I believe it would be unreasonable to ask the author's to setup such an experiment it might be nice to elaborate on this claim.

- The author's mention STL, although all the formulas are provided with LTL notation, it is unclear how STL is used here, is it used in the goal-oriented learning or something else?

---

> ### Author Rebuttal · Authors · 2025-07-30
>
> We greatly appreciate your thoughtful and positive review of our work. We have addressed the points you raised below and would be happy to provide any additional clarifications if needed. Thank you for your support.
>
> ## Weaknesses
>
> >ACQL is currently restricted to Deterministic Büchi Automaton (DBA).
>
> > How do the authors envision extending their approach to more general STL/LTL classes or more complex automata?
>
> We acknowledge that ACQL, like many automaton-based RL methods, currently focuses on tasks expressible by DBAs. However, we note that the majority of practically useful tasks, like those involving goal recurrence and safety, fall within this class. This is further supported by the fact that the most recent methods that are capable of handling more general specifications (e.g., those with persistence requirements) often do not demonstrate such tasks in their experiments [1]. This suggests the additional complexity they introduce remains a barrier in practice, and extending our approach to broader temporal logic classes remains a promising direction for future work.
>
> We expect the first step in extending our approach will, similar to other methods, involve using non-deterministic Good-for-MDPs automata [2] (e.g., Limit-Deterministic Büchi Automata) to define our product CMDP and augmenting the action-space so that the policy may choose how to optimally resolve the resulting non-deterministic choices [3]. We are also interested in investigating how these non-deterministic choices are resolved in the presence of other reward signals and constraints.
>
> > it is not clear what they mean by stationary safety constraints.
>
> > in theory one could apply CMDP methods to the product CMDP
>
> Thank you for raising this point. In this context, we mean by non-stationary that the safety constraints can change throughout a task as the agent progresses, as opposed to changing between learning and execution. It is possible to overcome this initial limitation by applying standard CMDP methods to the product CMDP, but we show in our second ablation (row 4 of Table 2) that standard sum-of-cost CMDP constraints struggle to address the safety requirements of our tasks compared to the minimum-safety formulation we propose.
>
> > the authors claim that the baselines "RMs and the LOF cannot be easily enhanced with high-performing GCRL and Safe RL techniques without adopting substantial modifications to their learning algorithms and product MDP formulation"
>
> Thank you for the question. To clarify, applying HER to CRM-RS is not directly feasible, as HER requires a goal-conditioned agent, whereas CRM-RS operates in a standard product MDP that lacks an explicit goal representation for relabeling. Introducing goal-conditioning in such a way that does not modify the original temporal logic task is non-trivial; in our case, we achieve this by encoding each automaton state with its associated set of goals.
>
> Similarly, LOF relies on a single product MDP and could, in principle, use HER to train a goal-conditioned policy in place of subgoal-specific options. However, this would then require adopting a different strategy for addressing the dynamic safety constraints imposed by the LTL specification, as each LOF option is trained both for reaching its associated goal and adhering to safety constraints relevant to that goal.
>
> > The author's mention STL, although all the formulas are provided with LTL notation, it is unclear how STL is used here
>
> Thank you for raising this point. We specifically use the quantitative semantics (i.e., the robustness function) provided by STL in lines 216 - 219 to define the safety feedback function $c^A$. The remaining formulae are still expressed in STL notation, but without explicitly writing the default timing parameters ($[0, \infty]$) for each temporal operator.
>
> ## Questions:
>
> > Could the authors compare ACQL conceptually to approaches that use shielding or model-checking at runtime (to prevent unsafe actions), or to other LTL-guided RL methods [6]?
>
> These comparisons could be made as well. To summarize, shielding and model-checking for RL are, relative to our method, significantly more based on classic formal synthesis and verification techniques, and they consequently are more dependent on discrete models of the environment dynamics. Shielding specifically uses reactive synthesis to obtain, from a safety LTL specification and an abstraction of the system dynamics, a reactive system that can be used to minimally modify an arbitrary RL agent so that unsafe actions are never chosen. Runtime model checking approaches in general also rely on environment models to verify the overall safety of a policy and continuously check that runtime behavior matches the prediction of the model used in verification in order to ensure the policy remains safe. In contrast, our method does not rely on a model and supports more than just the safety fragment of LTL specifications.
>
> Works such as [4] by Hasanbeig et al. on deep LTL RL are more comparable to our method, as they are model-free RL approaches relying on rewards from an automaton. However, we include the Reward Machines baseline as a similar algorithm that is slightly simpler and more appropriate for our fully-observable setting.
>
> > In practice does $\gamma_c \to 1$ introduce instability issues? How sensitive is ACQL to the scheduling of the safety discount factor and the learning rates on the two timescales? Could you explain how the contraction property can still be shown as $\gamma_c \to 1$?
>
> In practice, this does indeed introduce some instability at values of $\gamma$ very close to 1, as small inaccuracies due to approximating the safety value function can quickly propagate and worsen the quality of the rest of the value function. Fortunately, this can be resolved by capping the $\gamma_c$ value at a maximum value of $0.99$. Because $\gamma_c$ only approaches and never equals or exceeds 1, the contraction property shown in Theorem 1 of [16] holds throughout the execution of our algorithm. On the other timescales, the learning rates for $Q^c$ and $Q^r$ did not have to be tuned at all for our algorithm to succeed in practice.
>
> > Further, did you have to tune the rate at which $\gamma_c$ approaches 1, or did a simple schedule (e.g., linear annealing) work out-of-the-box?
>
> Very minor tuning was initially required so that $\gamma_c$ increased slowly enough so that an initial safety value function could be quickly learned, but after this a straightforward schedule based on exponential decay worked out-of-the-box.
>
> > Does the training process itself ever cause safety violations in the environment? If the agent takes an unsafe action, is the episode terminated or is it allowed to continue with a cost? In real-robot experiments, how were safety violations handled to protect the hardware?
>
> Our algorithm does require experiencing transitions that violate safety requirements in order to learn the safety value function, and so we do not terminate episodes upon safety violations. We note that a buffer can be instated for the safety requirements so that safety violations do not immediately result in harm to the robot.
>
> For our real-robot experiments, training was exclusively performed in simulation to the point that safety constraints were not encountered during deployment of the policy.
>
> > How can ACQL be adapted to partially observable settings?
>
> We wish to clarify that a partially observable setting in this context refers to a domain where the agent can only receive partial observations (e.g. images) or imperfect information of the underlying system state, but the task specification is still defined in terms of this underlying state.
>
> In order to extend ACQL to this setting, we envision first augmenting the network so that throughout execution the history of observations is processed to obtain a latent representation of the underlying state, with which a Markovian policy should be well-defined.  Second, because atomic propositions relevant to the task specification can not be directly measured but instead must be deduced from the history of observations, our extended method should assume access to or learn a probabilistic function, similar to the one used in [4], for evaluating propositions.
>
> Subsequently, our method could maintain a probability distribution over automaton states to represent the agent’s belief about its current progress and update this distribution based on the incoming observations. Periodically sampling from this distribution would then yield a concrete subgoal list, which will be relabelled based on the final achieved subgoals. We believe this to be a very interesting and challenging direction for future developments of our work and the field of policy learning methods using temporal logic.
>
> [1] Jackermeier, Mathias, and Alessandro Abate. "Deepltl: Learning to efficiently satisfy complex ltl specifications for multi-task rl." The Thirteenth International Conference on Learning Representations. 2025.
>
> [2] Hahn, Ernst Moritz, et al. "Good-for-MDPs automata for probabilistic analysis and reinforcement learning." International Conference on Tools and Algorithms for the Construction and Analysis of Systems. Cham: Springer International Publishing, 2020.
>
> [3] Hahn, Ernst Moritz, et al. "Omega-regular objectives in model-free reinforcement learning." International conference on tools and algorithms for the construction and analysis of systems. Cham: Springer International Publishing, 2019.
>
> [4] Hasanbeig, Mohammadhosein, et al. "Reinforcement learning for temporal logic control synthesis with probabilistic satisfaction guarantees." 2019 IEEE 58th conference on decision and control (CDC). IEEE, 2019.

---

### Official Review · Reviewer_58ip · 2025-07-01

**Clarity:** 2
**Significance:** 2
**Originality:** 2
**Rating:** 4
**Confidence:** 4

**Summary:**

The authors propose ACQL (Automaton-Constrained Reinforcement Learning), a RL algorithm leveraging LTL task specifications and their automaton representation to guide the learning of a safe policy while leveraging goal conditioned learning techniques. Specifically, the ACQL method converts a given LTL formula into an automaton, from which safety and liveness constraints are extracted. It distinguishes itself from previous similar methods by the following contributions:
1. The use of Hindsight Experience Replay to help deal with sparse rewards. ACQL computes subgoals based on the automaton structure. Satisfaction of any of the subgoal provides a learning signal.
2. Incorporation of safety constraints. The method learns whether a current policy will violate a safety constraint for each state-action pair. This function, in turn, prevents the agent from choosing undesirable actions.

**Questions:**

The authors state that the minimal safety constraint is better over the discounted sum of cost. Is this always the case? Is the approach only appropriate for hard safety?

Why would you think of comparing the method against offline policy learning? It is a fundamentally different problem?

The function that relates propositions set of propositions currently true in the environment, referred to [8] as labeling function might be undefined. Both baselines require such a construct. It seems related to the use of Signal Temporal Logic and function $\rho$, which is unmotivated. Why would one prefer to use it over the labelling function?

**Ethical Concerns:**

["NO or VERY MINOR ethics concerns only"]

**Final Justification:**

Based on the authors responses to my questions. I have reviewed my final score.

**Limitations:**

Yes, the authors have addressed the limitation of the proposed method.

**Paper Formatting Concerns:**

No concern

**Quality:**

2

**Strengths And Weaknesses:**

The paper has the following strengths and weaknesses:

Strengths:
+ The paper is well-written.
+ The motivation for safety incorporation is novel.
+ The use of HER with goals derived from an LTL specification is interesting and is a nice trick to help address the challenge around designing reward functions.
+ The theoretical analysis of the proposed method is sound and clearly detailed in the appendix
+ Evaluation is extensive, includes relevant ablations. The use of a real-world experiments is a nice touch and refreshing compared to typical simulation environments.

Weaknesses:
- While there is value in the proposed approach, in its current form the overall contribution is not substantial enough to constitute a full paper at a conference. Trying to address some of the limitations highlighted in the end would benefit the overall impact of the paper.
- An analysis of the effect of the complexity of the task (large automaton) on the sample complexity is missing. It is particularly important as the observation space is expanded with the set of sub-goals (potentially quite large in complex tasks).
- The authors state that the minimal safety constraint is better over the discounted sum of cost. Is this always the case? Is the approach only appropriate for hard safety? I think it should be clarified.
- Another issue is in their choice of baselines. The authors position the paper as combining GCRL and Safe RL, yet the authors decided to not use any of the Safe RL methods as baseline with the justification that "Safe RL methods typically address stationary safety constraints". Only one of the tasks presented in the Experimental section feature such non-stationarity.
- It is not entirely clear to me whether extending CRM-RS with HER will not achieve the same result.
- The baseline results seem incorrect. I find it hard to believe that RMs, especially on 1st and 2nd task in the table, almost never solve the tasks. They are designed for these kinds of problems.

Minor writing issues to fix in future versions:
1. wrong citation for GCRL [1]:
2. line 232: should it not be in $\mathcal{S}^A$
3. formula (3) - elements such as $r_t$ seem to come from the batch but it is not explicitly specified

---

> ### Author Rebuttal · Authors · 2025-07-30
>
> Thank you for taking the time to review our work and for your detailed feedback. We have addressed each of your comments and clarified the points that may have been unclear in the original submission. If our responses resolve your concerns, we would sincerely appreciate your consideration in updating your score. If there are any remaining questions, we would be glad to provide further clarification.
>
> ## Weaknesses
>
> > While there is value in the proposed approach, the overall contribution is not substantial enough to constitute a full paper at a conference. Trying to address some of the limitations highlighted in the end would benefit the overall impact of the paper.
>
> Thank you for recognizing the value of our approach and for raising this point.
> We want to emphasize that the contributions of our work lie not only in the technical components, such as the integration of HER with a goal-providing product MDP and Temporal Logic-derived minimum-safety constraints, but also in the accompanying theoretical analysis and practical performance.
>
> These elements together offer a principled and scalable framework for learning to satisfy singular temporal logic specifications in high-dimensional continuous state spaces.
> We agree that addressing the remaining limitations would represent an important next step. However, these challenges are not unique to our method and are common to many approaches in the field. Our work takes meaningful steps in this direction and, in our view, provides a valuable foundation that the community can build upon.
>
> > An analysis of the effect of the complexity of the task on the sample complexity is missing. It is particularly important as the observation space is expanded with the set of sub-goals.
>
> We would like to clarify that the observation space is not necessarily expanded as the number of subgoals increases. As described in lines 172-176, our algorithm augments the observation only with the goal corresponding to the next relevant automaton state. When the automaton includes branching, we concatenate all possible successor automaton states to capture the multiple paths forward. However, in the absence of branching, only a single goal is included. This design keeps the observation space compact in the majority of practical scenarios.
>
> While it is theoretically possible for the observation space to grow with automaton size in tasks with extreme branching, we believe this is rare in the types of robotic applications and temporal objectives we are targeting.
>
> We agree that a deeper investigation into the connection between automaton complexity and sample complexity would be valuable. However, given our current scope, we believe the most immediate contributions lie in demonstrating the practical feasibility of Temporal-Logic-based learning in continuous, high-dimensional environments.
>
> > The authors state that the minimal safety constraint is better over the discounted sum of cost.
>
> Thank you for the question. We believe that using the minimal safety constraint provides advantages over the discounted sum of costs in almost all cases. We interpret hard safety as referring to constraints that cannot tolerate any violation, while soft safety allows for partial violations. Our approach can accommodate both by adjusting the threshold $L$ in Equation 2, in much the same way as one would tune $L$ in Equation 1 for the discounted cost formulation.
>
> The key benefit of our approach lies in the learning feasibility. Training a network to directly classify safety (i.e., learning the sign of $\min_{t \in [0, \infty]} c^A_t$) is significantly easier and more stable than learning to predict a discounted sum of costs and then thresholding the result. For evidence, consider the lower-bounds on sample-complexity for learning a Probably-Approximately Correct binary classifier given in Theorem 6.8 of [1] and for a real-valued regression model given in Theorem 2.1 in [2]. Additionally, the threshold $L$ in Equation 2 has a more direct interpretation as the maximum allowable safety violation, rather than the total accumulated cost, which is harder to reason about in safety-critical applications.
>
> We will make sure to clarify these points in the final version of the paper.
>
> [1] Shalev-Shwartz, Shai, and Shai Ben-David. Understanding machine learning: From theory to algorithms. Cambridge university press, 2014.
>
> [2] Morgenstern, Jamie H., and Tim Roughgarden. "On the pseudo-dimension of nearly optimal auctions." Advances in Neural Information Processing Systems 28 (2015).
>
> > The authors decided to not use any of the Safe RL methods as baseline with the justification that "Safe RL methods typically address stationary safety constraints". Only one of the tasks features such non-stationarity.
>
> We agree that the current justification in the paper is not sufficiently clear, and we will revise the quoted sentence to provide a more explicit and well-motivated explanation.
>
> Specifically, while Safe RL methods typically focus on stationary constraints, their scope is also fundamentally different from ours. They are not designed to satisfy general Temporal Logic specifications.
>
> We will clarify this reasoning in the final version of the paper, around lines 268 to 270.
>
> > It is not entirely clear to me whether extending CRM-RS with HER will not achieve the same result.
>
> We would like to clarify that simply extending CRM-RS with HER would not achieve the same result as our proposed method. While it is possible in principle to augment CRM-RS with HER, doing so is not straightforward. HER requires a goal-conditioned agent, and adapting CRM-RS to support this would require explicitly constructing a goal-providing product MDP.
>
> Even if such a combination were implemented, it would still rely on rollout termination to enforce safety constraints. Our experimental results show that this approach is significantly less effective than the minimum-safety constraint we introduce.
>
>
> > It is hard to believe that RMs, especially on 1st and 2nd task in the table, almost never solve the tasks.
>
> While reward machines are indeed well-suited for representing temporal logic tasks, their effectiveness has primarily been demonstrated in relatively small or discrete environments where rewards are encountered frequently and value function learning progresses more easily. In contrast, the environments we consider are significantly more challenging. They involve high-dimensional continuous state spaces and require long sequences of actions to reach even a single reward, resulting in extremely sparse reward signals.
>
> In such settings, standard reward-machine-based methods struggle to make meaningful progress, as the agent rarely observes positive feedback. This motivates our proposed approach using goal relabeling and safety-aware learning. Our results reflect this difficulty and highlight the necessity of the design choices made in our method.
>
> ### Writing Issues
>
> > Wrong citation for GCRL [1]
>
> We believe the Universal Value Function Approximators work is an appropriate citation, as it represents a starting point for the field of deep goal-conditioned RL.
>
> > line 232: should it not be in $\mathcal{S}^A$
>
>
> > formula (3) - elements such as $r_t$ seem to come from the batch but it is not explicitly specified
>
> While the states $s_t^A$ in Line 232 are also members of $\mathcal{S}^A$, we meant by writing $s_t^A \in \mathcal{B}_i$ that these are the states included in the batch of sampled transitions $\mathcal{B}_i$ used for computing the loss in Equation 3. We will further clarify this notation near Equation 3 in the final version of the paper.
>
> ## Questions:
>
> > What would you think of comparing the method against offline policy learning? It is a fundamentally different problem?
>
> We agree that combining offline policy learning with temporal logic objectives is an interesting and promising direction for future research. For instance, one could imagine collecting offline data using policies that satisfy certain LTL specifications and then training new policies to generalize to unseen temporal tasks. However, this would involve a different problem formulation, with different assumptions, training regimes, and evaluation goals.
>
> Our focus in this work is on online policy learning with sparse feedback and safety constraints, where the agent must learn from limited, task-driven interaction in high-dimensional continuous spaces. Exploring how offline learning techniques could complement or extend our framework is a natural and exciting next step, but beyond the scope of the current study.
>
> > The labeling function might be undefined. The use of Signal Temporal Logic and $\rho$ is unmotivated.
>
> In our formalism, we chose to define the automaton transition function directly as $\delta: \mathcal{Q} \times \Sigma \to \mathcal{Q}$ with $\Sigma = \mathcal{S}$, rather than introducing an explicit labeling function. This can be equivalently viewed as the composition of a labeling function $L: \mathcal{S} \to 2^{AP}$ and a transition function defined over $2^{AP}$. While this choice simplifies notation in our setting, we agree that including the labeling function explicitly would improve consistency with standard treatments of automata-based RL and with prior work such as [8]. We are considering adding this to the final version of the paper for completeness and clarity.
>
> The use of Signal Temporal Logic (STL) in our framework is still compatible with a labeling function. Each atomic proposition $p \in AP$ is associated with a function over states, and the labeling can be constructed based on their truth values. The robustness function $\rho$ provided by STL offers an additional advantage by enabling a principled and differentiable way to define our safety constraint function $c^A$, as introduced in line 217.

---

> > ### Author Response · Authors · 2025-08-06
> >
> > Thank you again for your review. As the discussion session is ending soon, we just wanted to re-express our availability to give any further clarification on the points we've discussed. If our discussion so far has addressed your concerns, we would very much appreciate your consideration in updating your score. Let us know if there is anything else we can clarify.

---

> > > ### Comment · Area_Chair_xEW7 · 2025-08-08
> > >
> > > Dear Reviewer 58ip,
> > >
> > > The authors have provided detailed responses to your concerns. Could you please confirm whether your concerns have been fully addressed? Your participation in the discussion is important to ensure a fair evaluation.
> > >
> > > AC

---

### Official Review · Reviewer_X2Fi · 2025-07-03

**Clarity:** 4
**Significance:** 4
**Originality:** 4
**Rating:** 5
**Confidence:** 3

**Summary:**

The paper introduces Automaton Constrained Q-Learning (ACQL), a RL algorithm to learn policy that satisfies LTL objectives in continuous domains. The method addresses the challenges of sparse rewards and limited scalability by formulating an augmented constrained MDP that incorporates both safety and liveness constraints extracted from the automaton structure. ACQL leverages goal-conditioned Q-learning with Hindsight Experience Replay (HER) to densify rewards and introduces a minimum-safety constraint formulation to enforce safety without relying on brittle reward shaping or rollout termination.

**Questions:**

I do not have any specific questions. However, I would like the authors to comment on my concerns mentioned in weaknesses part.

**Ethical Concerns:**

["NO or VERY MINOR ethics concerns only"]

**Final Justification:**

The authors have addressed my minor concerns.

**Limitations:**

yes.

**Paper Formatting Concerns:**

none.

**Quality:**

4

**Strengths And Weaknesses:**

Strengths
=========

The paper makes important advancements in the area of learning policies to satisfy LTL specifications. In particular, it nicely combines the decomposition properties of DBA-expressible LTL formulas with CMDP formulation to enable learning LTL-compliant policies. Such a compositional approach is known to be non-trivial, in general. The idea is well motivated, formal proofs are sound, and results are promising. I believe it could have good impact in design of trustworthy RL agents.


Weaknesses
==========

I do not have major concerns.

1. Given the main content (my pdf doesn't have appendix), it's unclear whether experimental setup considers deterministic or stochastic environments.

2. In obtaining the Automaton... paragraph, it appears that proposition identification is manual. If so, how would misidentification of these subgoals affect the results?

---

> ### Author Rebuttal · Authors · 2025-07-30
>
> Thank you very much for taking the time to review our work and for your positive evaluation. We have addressed the points you raised below and would be happy to provide any additional clarifications if needed. We sincerely appreciate your feedback and support.
>
> ## Weaknesses
>
> > It's unclear whether experimental setup considers deterministic or stochastic environments.
>
> Thank you for highlighting this point. Our current experiments are indeed implemented using a deterministic simulator, and we agree that this detail should be clarified more explicitly in the final version of the paper. That said, we would like to emphasize that our algorithm is formulated without any assumption on deterministic dynamics and is fully compatible with stochastic environments.
>
> > In obtaining the Automaton... paragraph, it appears that proposition identification is manual. If so, how would misidentification of these subgoals affect the results?
>
> We confirm that identification of subgoal propositions is currently performed manually, as described in lines 176-178. However, this process is straightforward and involves selecting atomic propositions that correspond to goal-reaching behaviors specified in the given temporal logic formula.  These are typically the propositions that encode states the agent must eventually reach.
>
> While it is possible to automate this step in general, we found that manual identification is reliable and poses a minimal one-time overhead in practice. Misidentification would only occur in the case of a malformed or ambiguous specification, in which case any high-level reasoning algorithm would likely fail to generate meaningful goals. In our experiments, we encountered no such issues.

---

### Note · Authors · 2025-08-11

Thank you to the Area Chair and all the Reviewers for their thoughtful feedback and constructive suggestions, which have helped us improve our work. We would like to use this final comment to reaffirm our commitment to incorporating the improvements discussed during the review process into the final version of the paper.

## Planned Improvements
1. As requested by reviewers Lw5u and v818, we will add a discussion of related works employing shielding or runtime model-checking, and we will also clarify our work’s single-task focus in contrast to recent literature on multi-task Temporal Logic RL [1–7].
2. As pointed out by reviewers v818 and 58ip, we will explicitly introduce a state-labeling function for defining our product CMDP, to improve clarity and ensure consistency with the established literature.
3. We agree with reviewer v818 that including Algorithm 1 in the main text will aid readers, and we will incorporate it into the final version.
4. We will clarify several specific points in the final text: the interpretation of $G^i$ as a Cartesian product (line 164, v818), the origin of states $s_t^A$ (line 232, 58ip), the precise meaning of non-stationary safety constraints (Lw5u), and the scope of Safe-RL baselines compared to our method (v818).

We sincerely thank all Reviewers once again for their valuable feedback!

---

### Decision · Program_Chairs · 2025-09-17

**Decision:**

Accept (poster)

**Comment:**

The paper proposes Automaton-Constrained Q-Learning (ACQL), a reinforcement learning algorithm designed to satisfy Linear Temporal Logic (LTL) objectives in continuous domains. The key idea is to translate an LTL formula into a deterministic Büchi automaton (DBA) and then construct an augmented product constrained MDP (CMDP), in which the environment state is combined with the automaton state and automaton-derived subgoals. To address reward sparsity, ACQL employs goal-conditioned RL with Hindsight Experience Replay (HER), treating automaton subgoals as intermediate targets that densify learning signals. To address safety, the authors introduce a minimum-safety constraint over trajectories, rather than a discounted cumulative cost, which biases the learning toward avoiding unsafe actions in a more principled way. The paper provides a convergence proof, empirical validation on continuous-control domains, ablations, and a real-world robotic manipulation experiment where a 6-DOF robotic arm successfully executes LTL-specified tasks.

**Strengths**
ACQL makes a well-motivated and technically sound contribution by combining LTL-to-automaton translation, CMDP formalization, HER-based goal conditioning, and a trajectory-level safety constraint. This is a non-trivial synthesis of ideas from multiple areas. The use of HER with automaton-derived subgoals is a compelling way to mitigate reward sparsity, while the minimum-safety constraint is a conceptually strong alternative to standard cumulative costs. The theoretical contribution is also meaningful: the construction of the product CMDP and the convergence proof for the two-timescale Q-learning scheme are carefully developed. The experimental evaluation is convincing, with ablations clarifying the role of HER and safety constraints, and with validation extending beyond simulation benchmarks to a real robotic arm, which strengthens the practical relevance of the work.

**Weaknesses**
The paper's choice of baselines is a point of concern. It does not compare against several relevant LTL-based RL approaches—such as automata embeddings, experience replay method via counterfactual reasoning, and cycle experience replay—which are necessary for stronger positioning within the literature. The paper also lacks an analysis of how automaton complexity affects sample efficiency, a crucial factor for scaling to more complex specifications. Clarity can be improved in several places. First, the definition of subgoal lists is ambiguous, and the construction algorithm in the appendix appears somewhat ad hoc. It is not clear why the design is formalized as a list, especially since the representation passed to the neural network effectively encodes goals as Boolean combinations of subgoals via fuzzy logic. Second, the experiment does not clarify whether the approach naturally supports more general goal structures, such as conjunctions/disjunctions of goals or overlapping subgoals, which raises questions about extensibility. Lastly, the paper makes implicit assumptions about the goal space that are not formally stated. In particular, it assumes all propositions in $\text{AP}_\text{goals}$ are defined over the same set of state dimensions for goal-conditioned learning, which prevents handling cases where e.g. some propositions depend on positions while others depend on velocities.

During the rebuttal, the authors added a comparison with reward machines. However, this alone is insufficient; a broader set of relevant LTL-based RL baselines should be included for a fair and convincing evaluation. Overall, the paper presents a technically sound and well-motivated contribution at the intersection of LTL-based RL and safe RL, combining HER-based goal densification with a minimum-safety constraint in a product CMDP formulation. The meta-reviewer encourages the authors to improve the paper’s positioning within the literature and expand baseline coverage; in addition, the scalability and expressiveness limitations require clearer discussion. Addressing these points would significantly strengthen the work.